# Human cytomegalovirus in breast milk is associated with milk composition and the infant gut microbiome and growth

Kelsey E. Johnson[1] ✉, Nelmary Hernandez-Alvarado[2], Mark Blackstad[2], Timothy Heisel[2], Mattea Allert[1], David A. Fields[3], Elvira Isganaitis[4], Katherine M. Jacobs[5], Dan Knights[6,7], Eric F. Lock[8], Michael C. Rudolph[9], Cheryl A. Gale[2], Mark R. Schleiss[2], Frank W. Albert[1,12], Ellen W. Demerath[10,12] & Ran Blekhman[11,12]

Human cytomegalovirus (CMV) is a highly prevalent herpesvirus that is often transmitted to the neonate via breast milk. Postnatal CMV transmission can have negative health consequences for preterm and immunocompromised infants, but any effects on healthy term infants are thought to be benign. Furthermore, the impact of CMV on the composition of the hundreds of bioactive factors in human milk has not been tested. Here, we utilize a cohort of exclusively breastfeeding full-term mother-infant pairs to test for differences in the milk transcriptome and metabolome associated with CMV, and the impact of CMV in breast milk on the infant gut microbiome and infant growth. We find upregulation of the indoleamine 2,3-dioxygenase (IDO) tryptophan-to-kynurenine metabolic pathway in CMV+ milk samples, and that CMV+ milk is associated with decreased *Bifidobacterium* in the infant gut. Our data indicate two opposing CMV-associated effects on infant growth; with kynurenine positively correlated, and CMV viral load negatively correlated, with infant weight-for-length at 1 month of age. These results suggest CMV transmission, CMV-related changes in milk composition, or both may be modulators of full-term infant development.

Human cytomegalovirus (CMV) is a member of the herpesvirus family with a global seroprevalence of ~85% in women of childbearing age[1]. CMV is a double-stranded DNA virus that can infect multiple cell types, including epithelial, endothelial, and immune cells[2]. Initial infection in healthy individuals is often asymptomatic, followed by lifelong viral latency. The most common mode of CMV transmission in infants is through breast milk, as during lactation, CMV locally reactivates in the mammary gland in virtually all seropositive women[3–6]. Following mammary CMV reactivation, the presence of viral DNA in milk can be detected in both milk cells and whey[7–9].

Postnatal CMV transmission via breast milk is thought to be benign in full-term, non-immunocompromised infants[10]. However, for preterm infants, postnatal CMV can have serious clinical consequences, including sepsis, thrombocytopenia, and long-term neurodevelopmental impairment[10]. Among preterm and very low birth weight infants fed CMV+ breast milk, ~20% are estimated to acquire CMV[10,11]. The rate of transmission in full-term infants breastfed by seropositive mothers is estimated at up to 70%[12–14].

Despite the prevalence and clinical importance of mammary CMV reactivation, little is known about its relationship to human milk composition. CMV reactivation could lead to a local immune response and viral regulation of host metabolism that could impact milk composition. Conversely, differences in milk composition could modify the risk of CMV reactivation, also leading to associations between CMV

reactivation and milk composition. Associations between mammary CMV reactivation and the hundreds of nutritive and bioactive components of human milk have mostly not been assessed, but one study found an increase in pro-inflammatory cytokines in the setting of maternal CMV reactivation during lactation[15]. If CMV reactivation does alter human milk composition, it would be important to understand the impact of these changes on the infant. Variation in milk composition is associated with infant development, including the gut microbiome and immune system[16–18]. For preterm infants, who strongly benefit from human milk feeding[19], an understanding of CMV-related changes in milk composition and their impact on infant health outcomes is critical.

One approach to understanding the mechanism by which CMV affects host physiology is to quantify the host transcriptional response and the metabolome in the context of CMV infection. The impact of CMV on host gene expression has been examined in cultured cells[20–24] and in the blood of kidney transplant recipients[25], but not in the context of mammary reactivation. Similarly, the metabolome during CMV infection has been described in cultured cells[26,27] and infant urine[28], but not in milk. The milk transcriptome and metabolome provide complementary profiles of the physiology of the lactating mammary gland and milk composition[16,29–31]. Although the clinical impact of postnatal CMV transmission is far greater for preterm than for term infants, the mechanisms by which CMV alters or is altered by human milk composition can be studied using milk from term mother-infant dyads.

In this study, we aimed to identify differences in human milk composition and infant outcomes associated with CMV reactivation in a deeply phenotyped cohort of lactating mothers and their full-term infants. Leveraging multi-omics data from mother-infant dyads, we tested for differences in the milk transcriptome, milk metabolome, and infant fecal metagenome associated with milk CMV reactivation (Fig. 1). Further, we utilized anthropometric data to characterize differences in infant growth associated with milk CMV reactivation. Our results indicate that there are previously unappreciated differences in milk composition, infant gut microbiome composition, and growth in healthy full-term infants exposed to CMV through breast milk.

## Results

### Identifying CMV-positive samples from shotgun DNA sequencing of human milk

As CMV is a DNA virus, its presence can be detected in the lactating mammary gland by measuring CMV DNA in milk[8]. Viral shedding into breast milk typically begins within one week postpartum, and peaks 1–2 months postpartum[5]. We leveraged existing shotgun DNA sequencing data from 1-month postpartum milk samples[16] ($N = 276$) to identify milk samples with CMV viral shedding (Fig. 1A). We mapped milk-derived DNA sequencing reads to the CMV genome and designated any sample with at least one read mapped to the CMV genome as CMV+ (96/276, 35% CMV+; Fig. 1B, Supplementary Data 1). Hereafter, samples with no CMV-mapped reads were designated as CMV−. To ensure our results were not dependent on this choice of threshold, we repeated the main analyses in this manuscript using a series of higher thresholds for the required proportion of CMV-mapped reads to designate a sample as CMV+. We saw no qualitative difference in our results across the range of tested thresholds (Supplementary Data 2; but see infant growth section below). The mean proportion of CMV-mapped reads in samples designated as CMV+ was about 1 per 100,000 sequenced reads (Fig. 1C), reflecting the fact that the vast majority of DNA in these milk samples comes from human cells.

Milk DNA was extracted and sequenced using two approaches for two distinct original goals: low-pass human whole-genome sequencing (WGS) or shotgun metagenomic sequencing (SMS). The main difference between these approaches was the extraction protocol (see details in Methods). Within samples that had CMV-mapped reads from both datasets ($N = 24$), there was a positive correlation in the

proportion of CMV-mapped reads (Spearman's rho = 0.81, $P = 3.47 \times 10^{-5}$; Supplementary Fig. 1). Mapped reads were widely distributed across the CMV genome (Fig. 1D). There was no significant difference in the mean total read count for CMV+ vs. CMV− samples (two-sided $t$ test, $P = 0.74$; Supplementary Fig. 2), suggesting that read depth did not bias our approach to detect CMV+ samples. Within CMV + samples, there was no significant difference in the mean proportion of reads that mapped to the CMV genome between the two sources of DNA sequencing data (two-sided $t$ test, $P = 0.93$; Supplementary Fig. 3). Taken together, these results suggest that our detection of CMV+ samples was not biased by technical factors or sequencing pipeline.

To validate our approach of identifying CMV+ milk samples from shotgun sequencing data, we utilized qPCR to detect CMV DNA in a subset of 187 of the same milk samples by an established protocol[32,33]. The shotgun sequencing and qPCR results were in strong agreement (Fig. 1E, Supplementary Fig. 4, Supplementary Data 1). Taking qPCR detection of CMV as ground truth, the shotgun sequencing approach had 92.7% sensitivity (95% CI: 92.5–92.9%) and 94.7% specificity (95% C.I. 94.6–94.8%) to identify CMV+ samples (Supplementary Data 1). Conversely, taking the shotgun data as ground truth, qPCR detection had 87.9% sensitivity (95% CI: 87.7–88.2%) and 96.9% specificity (95% CI: 96.8–97.0%). Within milk samples identified as CMV+ by both approaches, the qPCR viral load estimate was highly correlated with the proportion of mapped reads from shotgun data (Pearson's $r = 0.88$, $P = 3.3 \times 10^{-17}$; Fig. 1F). As shotgun sequencing data was available for a larger sample, and all major results were consistent when using only the qPCR data (Supplementary Data 2), we moved forward with our CMV status designations from mapping shotgun reads to the CMV genome.

Comparing the maternal characteristics of CMV+ vs. CMV− milk samples, we observed that CMV+ milk samples were less likely to come from mothers who self-identified as white/European-American (74% in CMV+ vs. 91% in CMV−, $P = 3.1 \times 10^{-4}$, $q$ value = $3.7 \times 10^{-3}$, Fisher's exact test; Supplementary Data 3). This is consistent with previous epidemiological estimates that CMV seropositivity is higher in non-white than white populations worldwide[34–36]. All other tested maternal traits were not significantly different between CMV+ and CMV− groups ($q$ value > 0.25 for all other tests; Supplementary Data 3).

### Immune response genes are upregulated in CMV+ milk samples

Human milk contains RNA from the milk-producing mammary epithelial cells and immune cells[37–40]. Thus, gene expression analyses of human milk provide a profile of the lactating mammary gland[29,30]. Using RNA-sequencing data we previously generated from the same milk samples studied here ($N = 221$)[16], we tested for differential expression of 17,675 genes in CMV+ vs. CMV− milk samples (Fig. 2A). Maternal age, maternal pre-pregnancy BMI, maternal self-reported race, maternal parity, infant age in days, sample RIN, RNA-sequencing pool, and the mass RNA extracted from the sample were included as covariates. 36 genes were significantly differentially expressed ($q$ value < 0.05), 34 of which were upregulated in CMV+ milk (Fig. 2B, Supplementary Data 4). These 34 upregulated genes were enriched for pathways related to the immune response to viral infections (Supplementary Data 5), with "cellular response to interferon-gamma" as the most significant pathway (GO:0071346, odds ratio = 74.5, $P = 5.22 \times 10^{-15}$, $q$ value = $2.70 \times 10^{-12}$). Upregulation of interferon-stimulated genes is a typical feature of the immune response to CMV infection[22,41,42] (Supplementary Fig. 5). Within CMV+ milk samples, the proportion of CMV-mapped DNA reads and expression of the differentially expressed genes was significantly positively correlated for two genes: BATF2 and IDO1 ($q$ value < 0.05, Supplementary Data 6).

As our bulk milk RNA-sequencing data derives from all the cells in our milk samples, we leveraged publicly available single-cell RNA-sequencing data from human milk[38] to explore the expression patterns of the 36 differentially expressed genes across milk cell types. We observed that genes more highly expressed in our CMV+ milk samples

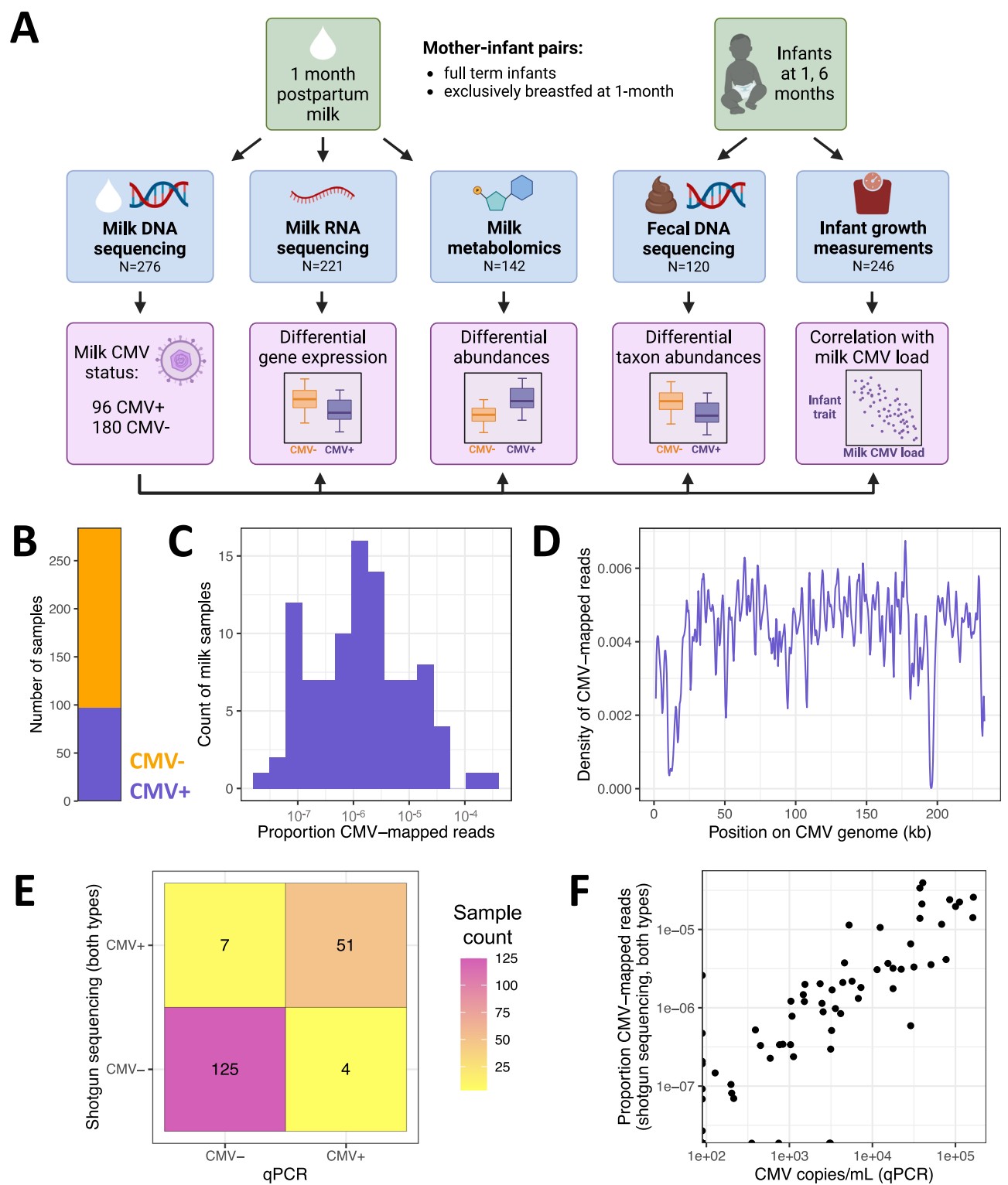

**Fig. 1 | Study overview and identifying CMV in human milk shotgun sequencing data. A** Study overview. **B** Count of milk samples identified as CMV+ ($N = 97$, purple) or CMV− ($N = 187$, orange). **C** The distribution of CMV-mapped DNA reads, as a proportion of all DNA reads, across milk samples that had at least one read mapped to the CMV genome. **D** Density of CMV-aligned reads across the CMV genome from all CMV+ milk samples. The density refers to the fraction of all CMV-mapped reads aligned to a particular region of the CMV genome. The density dips close to zero at repetitive regions in the CMV genome[97]. **E** Agreement of milk CMV status designations using shotgun DNA sequencing (*y* axis) vs. qPCR (*x* axis). **F** Within samples designated CMV+ by both qPCR and shotgun sequencing, the viral load estimates from the two methods were correlated (two-sided Pearson's $r = 0.88$, $P = 3.3 \times 10^{-17}$).

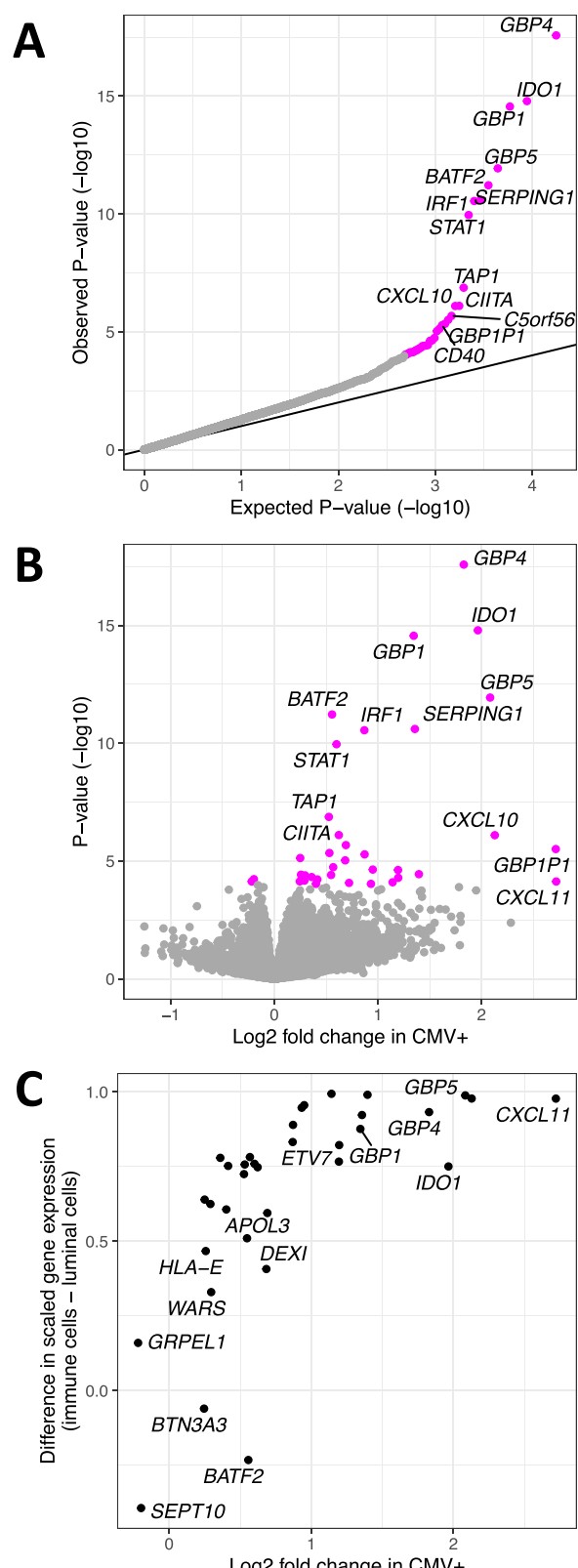

**Fig. 2 | Differences in gene expression associated with CMV+ human milk.**
Differential gene expression analysis comparing CMV− to CMV+ milk samples.
**A** QQ-plot from the results of differential gene expression analysis in DESeq2[49]. The
$x$ axis plots the expected $P$ value for the number of genes tested following a uniform
distribution of P values from 0 to 1, and the $y$ axis plots the observed two-sided
unadjusted $P$ values. Genes whose $P$ value was below the false discovery rate
threshold of 5% are colored in magenta. **B** Volcano plot comparing estimated effect
sizes of CMV+ on milk gene expression ($x$ axis) with each gene's unadjusted two-
sided $P$ value ($y$ axis). Genes whose $P$ value was below the false discovery rate
threshold of 5% are colored in magenta. P values and log2 fold change were cal-
culated in DESeq2[49]. **C** Comparison of log fold change in CMV+ samples from our
bulk RNA-seq data ($x$ axis) vs. gene expression in a publicly available human milk
single-cell RNA-seq dataset[36] ($y$ axis). Gene expression from milk single cells is
plotted as the difference between scaled gene expression in immune cells and
mammary luminal cells, to display that genes more highly expressed in our CMV+
milk samples tended to be more highly expressed in the immune cells in milk.

includes both CMV+ and CMV− samples. These results suggest that the
elevated expression of these genes in CMV+ milk samples stems from
an increased proportion of immune cells in CMV+ milk. This is
potentially consistent with previous studies showing an increase in T
cells in CMV+ human milk[42,43], though we only tested the estimated
proportion of all immune cells here due to the imprecision of cell-type
deconvolution of bulk RNA-seq data. We also note that the estimated
immune cell proportion is not independent of the differential gene
expression results above, as both analyses utilize the same bulk RNA-
sequencing data.

## Differentially abundant metabolites in CMV+ samples indicate higher activity of the IDO tryptophan-to-kynurenine metabolic pathway

The human milk metabolome reflects cellular processes in the mam-
mary gland and the composition of nutritive and bioactive compo-
nents delivered to the infant[44]. We tested for differential abundance of
458 metabolites between 58 CMV+ and 84 CMV− milk samples in a
regression model including study site, parity, maternal age, maternal
pre-pregnancy BMI, maternal self-identified race, maternal gestational
diabetes status, and maternal Healthy Eating Index score as covariates
(Supplementary Fig. 7, see Methods). Two metabolites were sig-
nificantly differentially abundant after correcting for multiple tests
($q$ value < 0.05, Supplementary Data 7): kynurenine (CMV+ estimated
effect = 0.74, $P = 2.3 \times 10^{-6}$, $q$ value = $1.2 \times 10^{-3}$; Fig. 3A) and its meta-
bolite kynurenic acid (CMV+ estimated effect = 0.79, $P = 5.8 \times 10^{-6}$,
$q$ value = $2.7 \times 10^{-3}$; Supplementary Fig. 8A).

The increased abundance of kynurenine and kynurenic acid in
CMV+ samples is concordant with the upregulation of the *IDO1* gene we
observed in our gene expression data (Fig. 3B). *IDO1* encodes indolea-
mine 2,3-dioxygenase (IDO), the rate-limiting enzyme in the tryptophan-
to-kynurenine metabolic pathway (Fig. 3C). The kynurenine/tryptophan
ratio was more significantly associated with CMV status than kynur-
enine alone (CMV+ estimated effect = 0.82, $P = 9.4 \times 10^{-7}$; Supplemen-
tary Fig. 8B). Within CMV+ milk samples, the kynurenine/tryptophan
ratio was positively correlated with the proportion of CMV-mapped
reads (Beta = 0.19, $P = 6.3 \times 10^{-3}$; Supplementary Fig. 8C). We did not
observe a difference in the abundance of tryptophan by CMV status
(CMV+ estimated effect = −0.23, $P = 0.20$, $q$ value = 0.85). Milk *IDO1*
expression was also positively correlated with the kynurenine/trypto-
phan ratio of abundances in milk, independent of milk CMV status
(Beta = 0.35, $P = 6.9 \times 10^{-8}$; Fig. 3D), illustrating the strong link between
expression of *IDO1* and the abundance of these metabolites.

## Milk CMV status is correlated with composition of the infant gut microbiome

Variation in human milk composition has been previously associated
with the infant gut microbiome[16,45–47]. Motivated by the differences in

tended to also be more highly expressed in immune cells in milk in the
single-cell data (Spearman's rho = 0.79, $P = 4.8 \times 10^{-7}$; Fig. 2C). CMV+
milk samples also had a higher estimated proportion of immune cells
(mean 16.5% in CMV+ vs. 12.6% in CMV−, $P = 0.041$, two-sided Wilcoxon
rank sum test; Supplementary Fig. 6; see Methods). We note that the
CMV status of the milk samples in the reference single-cell dataset
($N = 15$) is unknown, but given the high prevalence of CMV it likely

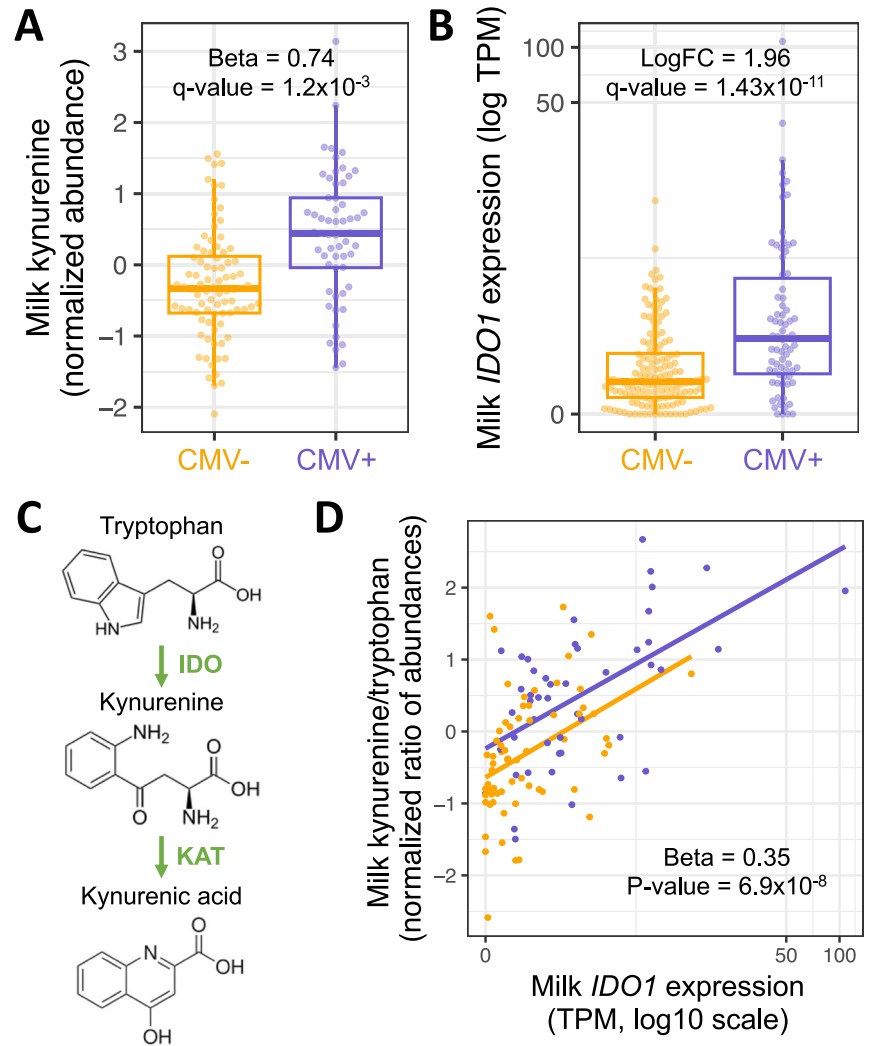

**Fig. 3 | The IDO tryptophan-to-kynurenine pathway is upregulated in CMV + milk. A** Kynurenine abundances in CMV− ($N = 84$, orange) vs. CMV+ ($N = 58$, purple) milk samples. Plotted kynurenine levels (y axis) are residuals after correcting for covariates included in the differential abundance analysis (see Methods). **B** *IDO1* expression in CMV− ($N = 147$, orange) vs. CMV+ ($N = 74$, purple) milk samples. Each dot represents a milk sample. LogFC: log fold-change between CMV+ and CMV− samples. In boxplots, the thick center line represents the median, the upper and lower hinges represent the 75th and 25th percentiles, and the whiskers extend to the largest/smallest value no further than 1.5 times the interquartile range from the hinge. **C** *IDO1* encodes the enzyme indoleamine 2,3-dioxygenase (IDO), which performs the rate-limiting step converting tryptophan-to-kynurenine. Kynurenic acid is metabolized from kynurenine by the KAT enzyme. **D** Correlation between *IDO1* expression (x axis) and the ratio of kynurenine and tryptophan abundances (y axis) in milk samples ($N = 111$), stratified by CMV status (orange: CMV−, purple: CMV+). Each dot represents a milk sample. The effect estimate and unadjusted two-sided *P* value were calculated in a linear regression of the log scaled kynurenine/tryptophan ratio residuals after regressing covariates (see Methods), against *IDO1* log TPM and milk CMV status.

milk composition we observed between CMV+ and CMV− milk samples, we next tested for associations between milk CMV status and composition of the infant gut microbiome. We previously generated shotgun metagenomic data from infant feces collected at 1 and 6 months postpartum ($N = 127$ mother/infant pairs at 1 month, $N = 120$ at 6 months)[16,48]. To explore a potential relationship between milk CMV status and the overall structure of the infant fecal microbiome, we first reduced the dimensionality of the microbial taxon abundance table using principal component analysis (each time-point analyzed separately, see Methods). We then tested for associations between milk CMV status and the microbial principal components (PCs). Milk CMV status was significantly correlated with PC3 of the 1-month infant fecal metagenomes (Beta = 1.79, $P = 1.1 \times 10^{-3}$, $q$ value = $5.6 \times 10^{-3}$; Fig. 4A, Supplementary Data 8). The top-loading taxa in 1-month PC3 were species of *Bifidobacterium* (negatively correlated with PC3; Supplementary Fig. 9). Milk CMV status was not associated with the 6-month taxon abundance PCs (Supplementary Data 8). Separately, we

performed principal component analysis on the microbial genetic pathway abundances estimated from shotgun metagenomic data, and milk CMV status was not associated with any of the pathway PCs (Supplementary Data 9). Milk CMV status was not associated with infant fecal alpha diversity at 1 month (Beta = 0.29, $P = 0.15$) or 6 months (Beta = 0.06, $P = 0.70$). Infant delivery mode (cesarean vs. vaginal), maternal parity, maternal age, maternal self-identified race, maternal pre-pregnancy BMI, maternal gestational diabetes status, maternal Group B streptococcus status, fecal sample collection site, and maternal Healthy Eating Index[49] were included as covariates in regression models comparing milk CMV status to the infant fecal microbiome. Two additional covariates were included in models with 6-month infant fecal samples: exclusive breastfeeding status and introduction of complementary foods (Methods).

Given the association between milk CMV status and composition of the infant fecal microbiome, as captured by taxon abundance PCs, we next tested for associations between milk CMV status and

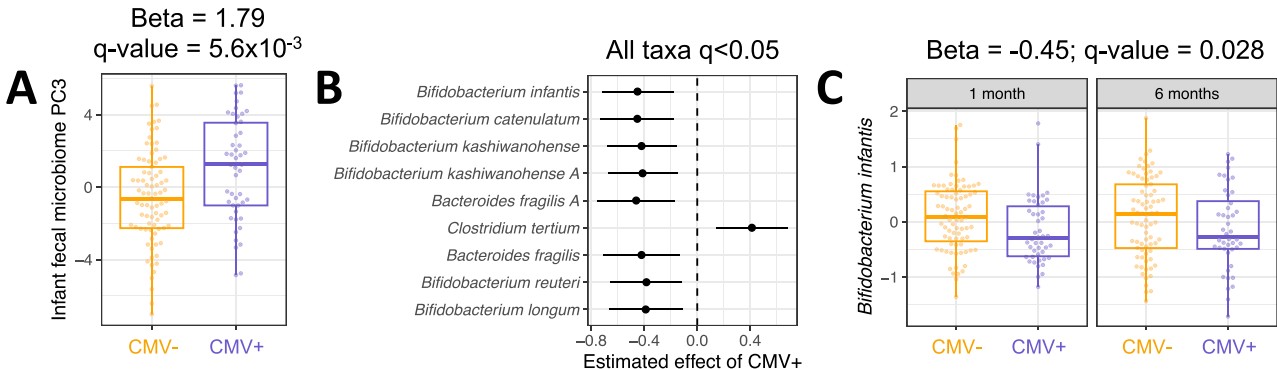

**Fig. 4 | Differences in gut microbiome associated with CMV+ milk.**
**A** Comparison of PC3 values for 1-month infant fecal samples fed CMV− ($N = 81$, orange) vs. CMV+ ($N = 48$, purple) breastmilk. Principal component analysis was performed on the taxon abundance table for infant fecal samples at 1 month of age. Each dot represents an infant fecal sample. Plotted PC3 levels are residuals after correcting for covariates included in the association analysis with milk CMV status (see Methods). **B** Estimated effect of CMV+ milk on normalized microbial taxa abundances in the infant gut, modeling samples from both 1 month ($N = 76$ CMV−, $N = 46$ CMV+) and 6 months ($N = 77$ CMV−, $N = 44$ CMV+) of age in a linear mixed model with infant age as a covariate (Methods). All taxa listed had a $P$ value below a false discovery rate of 5%. Taxa are arranged from smallest (top) to largest (bottom) $P$ value. Taxon names ending in 'A' were identified as distinct species by sequence identity in the reference genome database (see Methods). **C** The distribution of *Bifidobacterium infantis* abundances in the infant fecal microbiome, for infants fed CMV− (orange) or CMV+ (purple) milk, at 1 month ($N = 76$ CMV−, $N = 46$ CMV+) and 6 months ($N = 77$ CMV−, $N = 44$ CMV+) of age. Plotted *B. infantis* levels are residuals after correcting for covariates included in the association analysis with milk CMV status (see Methods). In **B** and **C**, taxon relative abundances were centered log-ratio transformed and scaled to mean 0, standard deviation 1 before association analysis. In boxplots, the thick center line represents the median, the upper and lower hinges represent the 75th and 25th percentiles, and the whiskers extend to the largest/smallest value no further than 1.5 times the interquartile range from the hinge.

abundances of individual microbial taxa. We modeled the abundances of 56 microbial species' in both 1 and 6 month old infants together in a linear mixed effects model (see Methods). Abundances of nine taxa were significantly correlated with milk CMV status ($q$ value < 0.05), including six species of *Bifidobacterium* that were less abundant in the gut metagenomes of infants fed CMV+ milk; *Clostridium tertium*, which was more abundant in infants fed CMV+ milk; and *Bacteroides fragilis*, which was less abundant in infants fed CMV+ milk (Fig. 4B, Supplementary Data 10). The taxon with the strongest association with milk CMV status was *Bifidobacterium infantis* (Beta = −0.45, $P = 1.4 \times 10^{-3}$, $q$ value = 0.028; Fig. 4C). We did not test for associations between CMV and individual microbial pathways due to the lack of correlation with microbial pathway PCs (Supplementary Data 9).

We next tested to see if CMV-associated microbial species in infants were correlated with CMV-associated changes in the milk metabolome or milk gene expression. Neither infant 1-month metagenome PC3 nor any CMV-associated microbial species was associated with milk kynurenine or the proportion of CMV-mapped reads (Supplementary Data 11). Several CMV-associated genes expressed in milk were correlated with the abundances of CMV-associated microbial taxa, but these milk gene – infant microbe associations were attenuated by adding milk CMV status as a covariate to the regression model (Supplementary Fig. 10, Supplementary Data 12). Thus, we found little evidence for CMV-related changes in milk composition leading to the observed differences in the infant fecal microbiome.

### Milk CMV viral load is correlated with infant growth
Finally, we tested if exposure to CMV+ milk was associated with infant growth, measured as weight-for-length Z-score (WLZ), a commonly used nutritional status metric to assess adequacy of weight relative to length and age in infants[50]. Infants fed CMV+ milk had on average approximately one-third of a Z-score greater weight-for-length at 1 month of age compared to infants fed CMV− milk (Beta = 0.38, $P = 0.011$, $N = 246$; Fig. 5A, Supplementary Data 13). Associations between infant growth metrics and milk CMV status were tested in a regression model including the same metric at birth, infant race (parental report), maternal pre-pregnancy BMI, maternal gestational diabetes status, household income, and infant delivery mode as covariates. This relationship between WLZ and 1 month milk CMV

status was not present at birth or at 6 months of age (Fig. 5A, Supplementary Fig. 11A). Infants fed CMV+ milk had somewhat lower mean length-for-age Z score at 1 month (Beta = −0.27, $P = 0.025$, Fig. 5A, Supplementary Fig. 11B), and no difference in weight-for-age Z score at 1 month (Beta = −0.012, $P = 0.89$, Fig. 5A, Supplementary Fig. 11C). These results indicate that infants fed CMV+ milk in the first month of life tended to have weight growth that exceeded their length growth in the first month. However, this difference did not persist to 6 months of age.

Within infants fed CMV+ milk, we observed a negative correlation between the proportion of CMV-mapped reads in milk and infant WLZ at 1 month (Beta = −0.20, $P = 1.1 \times 10^{-3}$, $N = 74$; Fig. 5B), the opposite direction of the relationship when comparing CMV− and CMV+ groups. This correlation between milk CMV viral load and infant WLZ did not persist to 6 month infants (Beta = $8.6 \times 10^{-4}$, $P = 0.99$). We also observed a positive correlation between the CMV-mapped read proportion in milk and infant length-for-age at one month (Beta = 0.12, $P = 0.042$; Fig. 5C), and no correlation with infant weight-for-age at one month (Beta = −0.035, $P = 0.46$; Supplementary Fig. 12). The relationship between milk CMV load and infant growth can be seen in our sensitivity analysis of escalating thresholds to designate milk samples as CMV+: as the threshold increases, only the milk samples with the highest proportion of CMV-mapped reads are designated CMV+, and the effect estimate of CMV+ milk on infant WLZ and milk CMV status reduces (Supplementary Data 2). These results suggest that a factor other than CMV viral load itself is driving the CMV group differences in WLZ at 1 month.

Hypothesizing that the relationship between CMV status and infant growth could be due to CMV-related differences in milk composition, we tested for a relationship between milk kynurenine abundance and infant 1-month WLZ. Kynurenine was positively correlated with WLZ (Beta = 0.19, $P = 3.9 \times 10^{-3}$; Supplementary Fig. 13), a relationship that persisted when milk CMV status was added as a covariate (Beta = 0.19, $P = 0.014$). Further, when testing the relationship between milk kynurenine and infant WLZ in CMV+ and CMV− groups separately, there was a positive correlation for both groups; though, it was only significant in the CMV− group (CMV+: Beta = 0.13, $P = 0.31$; CMV−: Beta = 0.23, $P = 0.024$; Fig. 5D). Within infants fed CMV+ milk, when including both milk kynurenine and the proportion of CMV-mapped

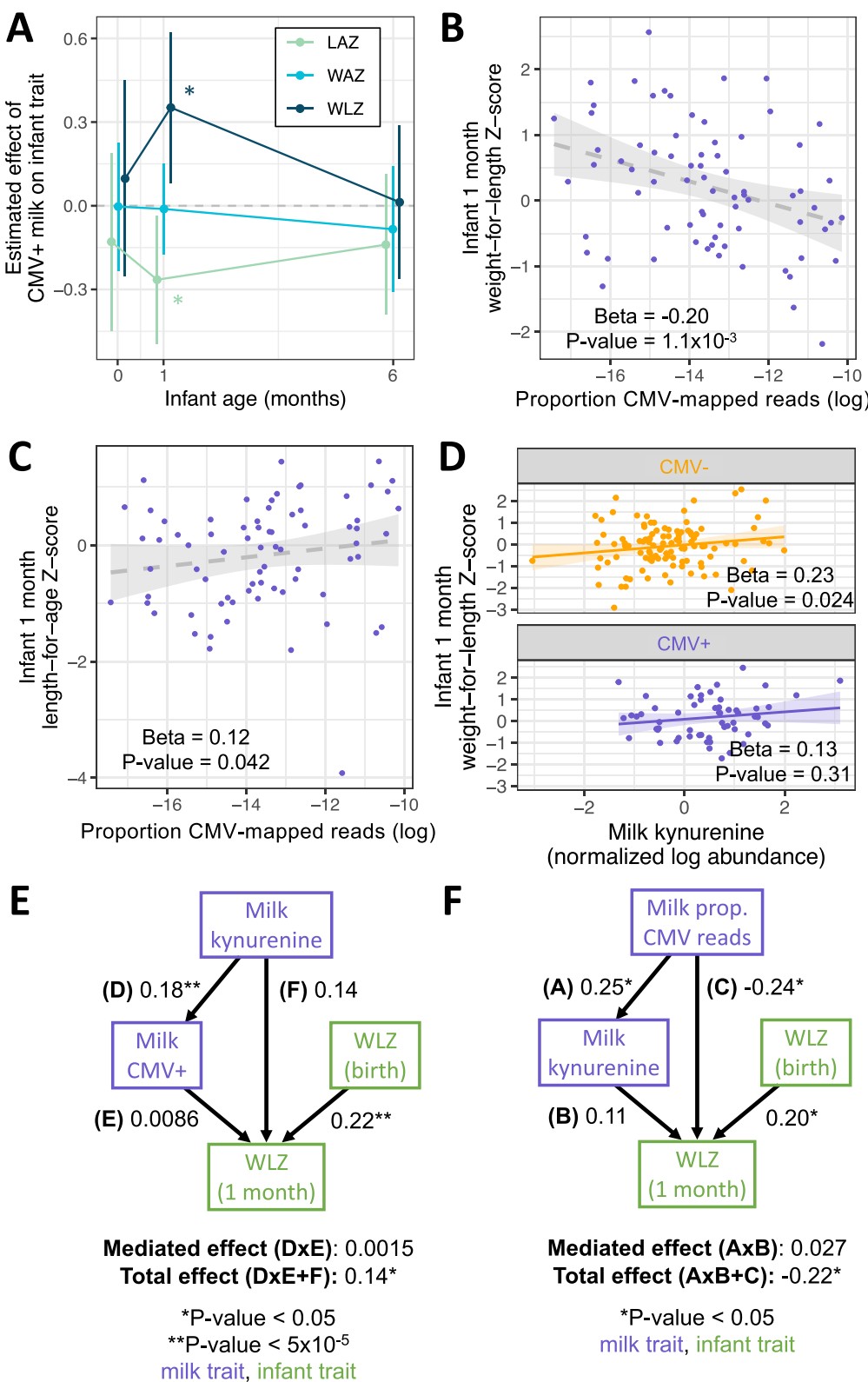

reads in milk, both terms were correlated with infant WLZ in opposing directions (kynurenine: beta = 0.22, $P$ = 0.088; CMV read proportion: Beta = −0.14, $P$ = 0.013). 1-month WLZ was not correlated with PC3 of the infant fecal microbiome nor with any of the microbial species associated with milk CMV status (Supplementary Data 11). Given that (1) accounting for milk kynurenine levels removes the association between milk CMV status and infant WLZ at one month; and (2) CMV

viral load is correlated with WLZ in the opposite direction as milk CMV status, even when including milk kynurenine levels; we conclude that increased kynurenine in CMV+ milk samples, or a correlated factor, is responsible for the positive association between milk CMV status and infant WLZ at 1 month.

We further investigated these relationships through structural equation modeling (SEM, Methods). First, when examining milk CMV

**Fig. 5 | Associations between CMV+ milk and infant growth.** Results of multivariate regressions of infant anthropometric measurements vs. milk CMV status, proportion CMV-mapped reads in milk, or milk kynurenine. All regression models included the equivalent *Z* score at birth as a covariate (except when the *Z* score at birth was the response variable). **A** Estimated effect of CMV+ milk on infant growth metrics at birth, 1 month, and 6 months of age. Error bars represent 95% confidence intervals. *$P < 0.05$; LAZ length-for-age Z score, light green; WAZ weight-for-age *Z* score, light blue; WLZ weight-for-length Z score, dark blue. Error bars represent a 95% confidence interval for the effect estimate. **B** Within infants fed CMV+ milk, there was a negative correlation between the proportion of CMV-mapped reads and infant WLZ at 1 month of age. The shaded gray area is the linear regression 95% confidence interval. **C** Within infants fed CMV+ milk, there was a positive correlation between the proportion of CMV-mapped reads and infant WLZ at 1 month of age. The shaded gray area is the linear regression 95% confidence interval. **D** There was a positive correlation between the abundance of kynurenine in milk and infant WLZ at 1 month, when tested for infants fed CMV+ ($N = 68$, purple) or CMV− ($N = 118$, orange) milk separately. The shaded areas are the linear regression 95% confidence intervals. All plotted infant growth metrics in **B**−**D** are residuals after correcting for covariates included in the association analyses with milk CMV status (see Methods). **E** Structural equation modeling of the relationship between milk kynurenine, milk CMV status, and infant 1-month WLZ. Multiple models were tested (Supplementary Fig. 14), with the best-fit model plotted here. Arrows next to numbers represent the standardized effect estimates, with asterisks indicating unadjusted two-sided *P* values. There was no evidence of milk kynurenine mediating a relationship between milk CMV status and infant 1-month WLZ, nor CMV status mediating a relationship between milk kynurenine and infant 1-month WLZ. **F** A structural equation model examining the relationships between milk proportion of CMV-mapped reads ('Milk prop. CMV reads', a proxy for viral load, within CMV+ milk samples), milk kynurenine, and infant 1-month WLZ. Arrows next to numbers represent the standardized effect estimates, with asterisks indicating unadjusted two-sided *P* values. The best-fit model plotted here found did not find evidence for kynurenine mediating the relationship between viral load and 1-month WLZ, but did support a direct effect from viral load to 1-month WLZ. **E**, **F** All tested models, their fit measures, and exact *P* values are shown in Supplementary Fig. 15. Purple boxes indicate milk traits, green boxes infant traits.

status, milk kynurenine, and infant 1-month WLZ, the best-fit model included no significant relationship between milk CMV status and infant WLZ (Fig. 5E). This model was chosen over a model with kynurenine mediating a relationship between milk CMV status and 1-month WLZ (Model 1 in Supplementary Fig. 13). We do not interpret the direction of the relationship between milk kynurenine and milk CMV status in these models, but rather that they support our above conclusion that the correlation between milk CMV status and 1-month WLZ is spurious. Additionally, when modeling the relationships between the proportion of CMV-mapped reads in CMV+ samples, milk kynurenine, and infant 1-month WLZ, the best-fit model included a direct effect from CMV read proportion to 1-month WLZ and no mediated effect through milk kynurenine (Fig. 5F, Supplementary Fig. 15). Thus, overall SEM supports both a positive relationship between milk kynurenine abundance and infant 1-month WLZ, and a negative relationship between CMV viral load and 1-month WLZ within babies fed CMV+ milk.

## Discussion

In this study, we found that the presence of CMV DNA in human milk is associated with milk gene expression and metabolite abundances, altered composition of the infant gut microbiome, and potential disruptions to infant growth in the first month of life. Notably, our study utilized a cohort of healthy, full-term infants in the United States; a population where the impact of CMV presence in breast milk or postnatal CMV transmission was largely thought to be negligible.

We utilized shotgun DNA sequencing from the cell pellet of human milk to identify samples with the presence of CMV at 1 month postpartum, and validated by qPCR. Our study demonstrates that nontargeted DNA sequencing of human milk can be used to identify CMV+ samples. We identified CMV DNA in 35% of 1 month milk samples by this approach, which is lower than the estimated seroprevalence for US women of childbearing age (~60%, but highly variable by geographic location and demography)[34,51,52]. As serum samples were not collected, the seropositivity rate of our study is unknown; however, the demographic characteristics of our cohort (mostly white and highly educated; Supplementary Data 3) indicate the rate is likely lower than the national average[34,51,52]. Virtually all seropositive women will have CMV reactivation in the mammary gland during lactation[53], and CMV viral loads are estimated to peak ~4–6 weeks postpartum[5,8], the approximate time of sampling for this study. While we may have been unable to detect CMV in some samples with a low viral load, the strong agreement between our shotgun sequencing and qPCR approaches suggests we robustly identified those samples with detectable CMV DNA. We also acknowledge that while viral reactivation during lactation is likely the primary cause of CMV DNA in breast milk, CMV could also be shed through breast milk in the context of primary infections or re-infections occurring late in gestation.

Using complementary milk RNA-sequencing and metabolomics approaches, we identified an upregulation of the *IDO1* tryptophan-to-kynurenine metabolic pathway in CMV+ milk samples. This pathway has previously been implicated in the immune response to CMV in studies of human cells and primary tissues[54,55], suggesting this association may be a response to mammary CMV reactivation. Additionally, one study found that providing kynurenine to human fibroblasts promoted CMV replication, and blocking *IDO1* decreased CMV replication[56]. Given our observational study design, we cannot determine if the association with increased *IDO1*/kynurenine is a cause or consequence of mammary CMV reactivation. Overall, the impact of CMV on milk composition was notably narrow, with a handful of genes and two metabolites differentially abundant between CMV+ and CMV− milk samples.

Under conditions of chronic viral infection, activation of the IDO pathway can lead to a more tolerogenic immune state[54], but the impact of elevated milk kynurenine and its metabolites on the infant is unknown. Kynurenine induction of the aryl hydrocarbon receptor (AHR) can cause immunosuppression via generation of regulatory T cells[57], and AHR activation may protect against necrotizing enterocolitis and inflammation in the infant gut[58,59]. Whether kynurenine metabolites in milk are at high enough concentrations to have physiological effects in the infant, and the potential impacts of CMV on this pathway, are possible areas of future investigation. We observed a positive association between milk kynurenine and infant growth at 1 month, with higher milk kynurenine correlated with lower length-for-age and greater WLZ, suggesting milk kynurenine levels might impact growth in early life independent of CMV status. It is important to note that while the impact of kynurenine on weight-for-length was of moderate effect statistically, differences of this magnitude are not generally of clinical significance for healthy term infants.

We also observed that within infants fed CMV+ milk, higher CMV-mapped read proportion (as a proxy for viral load) was negatively correlated with infant weight-for-length and positively correlated with length-for-age at 1 month of age. The total effect of CMV viral load on 1-month WLZ was estimated at −0.22, of comparable magnitude to the effect of WLZ at birth (0.20; Fig. 5F) but likely not of clinical significance in healthy infants. The associations between milk kynurenine or viral load (measured at 1 month) and infant growth did not persist to 6 months, indicating it did not have long-lasting effect. Previous research on the impact of postnatal CMV transmission on infant growth has primarily focused on two contexts: (1) very low birth weight infants in the NICU setting, and (2) in perinatally HIV-exposed but uninfected infants. Studies focused on very low birth weight infants

have found mixed evidence for impacts of postnatal CMV on anthropometric measures[60]. The largest study to date in very low birth weight infants found that postnatal CMV acquisition was associated with lower weight-for-age Z score at discharge, but no difference in length-for-age in a U.S. population[61]. In HIV-exposed but uninfected Malawian infants, breast milk CMV DNA load was negatively correlated with infant length-for-age and weight-for-age at 6 months (infant CMV status was unknown in this study)[62]. In addition, a study of Zambian infants found that postnatal CMV acquisition was associated with lower length-for-age Z score at 18 months in both HIV-exposed and HIV-unexposed infants, but no difference in weight-for-age by CMV status[63]. The context of our cohort is quite different from these previous analyses, yet cumulatively, these studies suggest that postnatal exposure and/or acquisition of CMV can impact infant growth.

We observed that exposure to CMV+ milk was associated with the composition of the infant gut microbiome in our cohort of breastfed babies. Specifically, CMV+ milk-exposed infants had lower abundances of *Bifidobacterium* species and higher abundances of *Clostridium tertium*. Lower *Bifidobacterium* abundance, particularly *B. infantis*, in the infant gut microbiome is associated with adverse health outcomes[64-67], though from this study we cannot determine if CMV-associated differences in the infant microbiome have any long-term health impact. *C. tertium* has been reported as potentially pathogenic in the infant gut[68,69]. Notably, milk kynurenine was not associated with the infant gut microbiome in our study, indicating that the potential effects of CMV viral load on infant growth and the infant gut microbiome may act through distinct pathways. A previous study by Sbihi et al. examined the impact of CMV acquisition on the infant gut microbiome. In a population-based birth cohort, early CMV acquisition (in the first 3 months of life) but not later CMV acquisition (between 3–12 months) was associated with lower alpha diversity (i.e. within-sample diversity)[70]. While our study is not directly comparable as we do not know infant CMV status, we did not observe a significant difference in alpha diversity in infants exposed vs. unexposed to CMV via milk. Sbihi et al. also observed increased incidence of childhood atopy with early CMV acquisition[70], a phenotype not currently assessed in our cohort.

A limitation of our study is the unknown serostatus of the infants at birth and subsequent timepoints, as infant blood samples were not available. As previous studies estimate up to 70% of breastfed term babies of seropositive mothers acquire CMV postnatally[12-14], it is possible that a substantial fraction of the babies fed CMV+ milk in our study had postnatally acquired CMV by 1 month of age. The infants in our study were also not tested for congenital CMV, which has a prevalence of ~4.5 per 1000 births in the US[33,71] and is often asymptomatic and undetected[72]. Further studies are required to characterize the impacts of CMV+ milk on growth and the gut microbiome in infants with and without CMV transmission, including in vulnerable preterm infants who most strongly benefit from receipt of human milk but also are at risk for short and long-term health complications upon acquisition of CMV by this route[3,61]. We cannot infer causality of the effects of CMV in milk on infant traits due to our observational study design, but these patterns can now be further explored and tested. Longitudinal sample collection and tracking of viral transmission would give further insight into the dynamics of CMV in human milk and corresponding impacts on the developing infant gut microbiome and immune system.

While there is growing awareness and understanding of the negative impacts of breastmilk-acquired CMV in preterm infants[10], it is generally thought to be benign in healthy term infants. Some have even speculated that there may be an evolutionary advantage to postnatal CMV acquisition, in the form of a 'natural immunization' or other immune-boosting effect for the infant[12]. We find that exposure to CMV+ milk is associated with reduction in beneficial microbes in the infant gut. Given the high prevalence of CMV globally, impacts on infant microbiome development could have a substantial impact at the population level. This study highlights not only these CMV-related changes but also, more generally, how 'normal' variation in human milk impacts healthy infant development.

## Methods

### Description of study population
This study made use of existing data from the Mothers and Infants LinKed for Healthy Growth (MILK) study. Recruitment protocols and study characteristics have been extensively described[16,48,73-76]. This study recruited gestating mothers intending to exclusively breastfeed their infants prenatally, and thus all adult participants were of female sex. Study visits occurred at two sites: the University of Minnesota (MN) and the University of Oklahoma Health Sciences Center (OK). All included infants were born at full-term. All participants provided written informed consent and the study protocols were approved by institutional review boards at the University of Minnesota, HealthPartners Institute for Education and Research, and the University of Oklahoma Health Sciences Center. The MILK study is registered with clinicaltrials.gov (NCT03301753).

The milk samples utilized in this manuscript were collected during a study visit ~1 month postpartum via a full breast milk expression two hours after a complete infant feed. Expressed milk volume and weight was recorded, milk was gently mixed, aliquots were made, and then stored at −80 °C within 20 minutes of collection and kept at −80 °C until thawed for RNA/DNA extraction or metabolomics analysis.

### Statistics and reproducibility
The genomics analyses described in this study were not pre-registered. Samples were collected and data generated prior to this observational study, and no statistical method was used to predetermine sample size. Two samples were excluded from the RNA-seq dataset due to having fewer than 10,000 genes detected; otherwise, all samples with data for each analysis were included.

### Human milk RNA extraction, sequencing, and gene expression quantification
The human milk RNA extraction protocol, sequencing, and gene expression quantifications used in this study have been previously described[16]. RNA extraction, library preparation, and sequencing were performed at the University of Minnesota Genomics Center (UMGC). Briefly, bulk RNA was extracted from the whole milk cell pellet to profile the gene expression of all cell types present in the milk sample. RNA was extracted from the cell pellet using the RNeasy Plus Universal HTP following the manufacturer's instructions. RNA libraries were prepared with the TakaraBio Stranded Total RNA Pico Mammalian kit and sequenced on an Illumina NovaSeq 6000 S2 flow cell with 2x150 paired-end reads in two pools. Gene-level quantifications were generated using RNA-SeQC v2.3.4[77].

### Analyses with publicly available single-cell RNA-seq data from human milk
Raw gene counts (MIT_Milk_Study_Raw_counts.txt.gz) and metadata (MIT_milk_study_metadata.csv.gz) were downloaded for the Nyquist et al. study[38] from the Broad Insitute Single Cell Portal (https://singlecell.broadinstitute.org/single_cell/study/SCP1671/cellular-and-transcriptional-diversity-over-the-course-of-human-lactation) on 6/3/2022. Gene counts for each cell were scaled to $\log(x/s + 1)$, where $x$ was the gene count in a cell and $s$ was a scaling factor. $s$ was calculated as the total counts per cell divided by the mean of total counts across all cells. For each of the 36 differentially expressed genes in our CMV+ milk samples, the scaled expression for each cell type was calculated as the mean scaled expression across all cells of that cell type, divided by the gene's mean scaled expression in the cell type with the highest mean expression. In Fig. 2C, immune cell expression included six cell types (T cells, eosinophils, dendritic cells, B cells, neutrophils,

macrophages), and mammary luminal cell expression included two cell types (luminal cell 1 and luminal cell 2).

Cell type proportions were estimated for each milk sample with bulk RNA-sequencing data as previously described[16], using a publicly available single-cell RNA-sequencing dataset from human milk[38] cells and Bisque[78]. Proportions of 8 cell types were estimated: two types of mammary epithelial cells (luminal cell 1, luminal cell 2) and six immune cell types (T cells, eosinophils, dendritic cells, B cells, neutrophils, macrophages). The estimated immune cell proportion was calculated as the sum of the six immune cell types.

## Human milk DNA extraction and sequencing
DNA was extracted and sequenced from human milk using separate protocols for different initial applications:

1. **Human low-pass WGS**: The DNA extraction protocol and sequencing for this application has been previously described[16]. In brief, DNA was extracted from the cell pellet at UMGC with the QIAamp 96 DNA Blood Kit, and sequenced by Gencove, Inc. for target sequencing depth of ~1x for the human genome.

2. **SMS**: DNA extraction and sequencing from milk samples for this application has also been previously described[16,48]. DNA was extracted using the PowerSoil kit, libraries constructed for metagenomics sequencing using the Illumina Nextera XT kit, and sequenced on an Illumina NovaSeq system using the S4 flow cell with the 2x150 bp paired-end V4 chemistry kit at UMGC.

## Identification of CMV-positive milk samples
We mapped DNA sequencing reads generated from human milk with the above two approaches to the human cytomegalovirus genome to identify milk samples with CMV DNA. Starting with the WGS DNA reads, we mapped the reads from each milk sample to seven CMV genome isolates from human milk[79] accessed from NCBI Genbank (https://www.ncbi.nlm.nih.gov/genbank/, MW528458–MW528464) using Bowtie2 v2.2.4[80]. Finding that the number of aligned reads across reference CMV isolates was in strong agreement, we continued with the aligned read count for each sample from isolate BM1 (accession MW528458) for reads from both WGS and SMS. We called milk samples CMV+ if they had at least one concordantly mapped read pair with MAPQ > 5 from either WGS or SMS. Of the 276 milk samples utilized in this study, 86 had both WGS and SMS (n = 34 CMV+), 132 only had WGS (n = 40 CMV+), and 58 had only SMS (n = 22 CMV+). The proportion of CMV-mapped reads was calculated for each CMV+ sample as the number of reads mapped to the CMV genome divided by the total number of sequencing reads, with counts from SMS and WGS data summed if both were available.

## PCR detection of CMV
qPCR reactions were performed starting with previously frozen whole milk samples in the Schleiss laboratory at the University of Minnesota. qPCR was performed on separate aliquots from 187 of the milk samples used for RNA and DNA sequencing above. 100 μL of thawed milk was extracted on the QIAcube (Qiagen, Hilden, Germany) with the DNeasy Blood and Tissue kit. Extracted samples were eluted in 100 μL of PCR-grade water and stored at −20 °C.

Two PCR reactions were performed with each milk sample elution, targeting (1) the UL83 gene[32] or (2) the immediate early gene 2 exon 5 region[33,81]. The viral load for each sample was reported as the average of the two qPCR reactions, and CMV+ status was designated for each sample with non-zero CMV detection from either reaction.

(1) *UL83 gene*: Multiplex qPCR was performed on milk sample eluate as previously described[32] in 25 μL total volume with 10 μL of template using LightCycler 480 Probes Master Mix (Roche, Basel, Switzerland) containing FastStart Taq DNA Polymerase, reaction buffer, dNTPs mix (with dUTP instead of dTTP), and MgCl2; as well as 0.4 μM primers, 0.1 μM probes, and 0.4 U/μL of uracil-DNA

glycosylase (UNG). PCR was performed using the Lightcycler 480 (Roche) under these conditions: 40 °C for 10 min, 95 °C for 10 min, followed by 45 cycles of 95 °C for 10 s and 60 °C for 45 s, then a final hold step at 40 °C. The primers used for UL83 were: forward primer, GGACACAACACCGTAAAGC; reverse primer, GTCAGCGTTCGTGTTTCCCA; and probe, CFR610-CCCGCAA CCCGCAACCCTTCAT-BHQ2. The primers used for NRAS were: forward primer, GCCAACAAGGACAGTTGATACAAA; reverse primer, GGCTGAGGTTTCAATGAATGGAA; and probe, FAM-ACAAGCCCACGAACTGGCCAAGA-BHQ1. A standard curve for the UL83 control was generated using 10-fold dilutions from $10^6$ to 10 copies/μL of a plasmid (UL83 fragment cloned in pCR2.1, using primers UL83_TM857F and UL83_TM1138R). The standard curve for NRAS was generated using five 10-fold dilutions starting with 200,000 to 20 pg/μL of human genomic DNA. The standard curve for NRAS PCR was generated using five 10-fold dilutions starting with 200,000 to 20 pg/μL of human genomic DNA (Roche).

(2) *Immediate early gene*: qPCR was performed as previously described[33] using the sample eluate described above. qPCR was performed using PerfeCTa Fastmix II master mix (QuantaBio). The CMV Taqman probe MGBNFQ (ThermoFisher) and primers (Integrated DNA Technologies) targeted the viral immediate early gene 2 exon 5 region[33,81]. The primers used were: forward primer, GAGCCGACTTTACCATCCA; reverse primer, CAGCCGGCGG-TATCGA; and probe, FAM-ACCGCAACAAGATT-MGBNFQ. Specimens were tested in triplicate using the Lightcycler 480 (Roche), and results were interpreted as described above. The sensitivity of the assay was 5 or fewer copies CMV/PCR reaction.

## Identification of differentially expressed genes by milk CMV status
Differential gene expression analysis between CMV− and CMV+ milk samples was performed in DESeq2[49] using the gene-level read count matrix generated with RNA-SeQC[77]. 17,675 genes were included in differential gene expression analysis. Maternal age, maternal pre-pregnancy BMI, maternal self-reported race, maternal parity, infant age in days, sample RIN, RNA-sequencing pool, and the mass RNA extracted from the sample were included as covariates. None of the individuals with transcriptomic data had gestational diabetes, so this was not included as a covariate. *P* values were adjusted for multiple tests using the default Benjamini and Hochberg method in DESeq2 v1.30.0[49,82]. Enrichment analysis of upregulated genes was performed with EnrichR v3.2[83], using "GO_Biological_Process_2021" as the reference gene ontology. To test for a correlation between CMV-mapped read proportion and gene expression, the same DESeq2 model was used, replacing CMV status with the CMV-mapped read proportion (logged and scaled to mean 0, s.d. 1) and including only CMV+ samples.

## Human milk metabolomics and identification of differentially abundant metabolites
Samples for milk metabolomics were prepared and analyzed as previously described[84] from frozen milk samples at BPGbio Laboratory (Framingham, MA). 200 μL aliquots of 1-month milk samples were thawed on ice, vortexed, and incubated in extraction solvent at a 1:2 sample:solvent volume ratio −20 °C for 30 minutes. The extraction solvent was a mixture of isopropanol, acetonitrile, and water at a ratio of 3:3:2. Metabolites were processed by three analysis techniques: gas chromatography combined with high-resolution TOF MS (GC-MS), reversed-phase liquid chromatography coupled with high-resolution MS (LC-MS), and hydrophilic interaction chromatography with tandem mass spectrometry (LC-MS/MS). A quality control sample containing a standard mixture of amino and organic acids purchased from Sigma-Aldrich as certified reference material, was injected daily to perform analytical system suitability test and monitor recorded signals for

day-to-day reproducibility. Samples were processed in 10 batches of 35 samples each, with 10 pooled milk samples and 40 external standards included to assess batch-to-batch variability.

Details for each technique were as follows. GC-MS: 75 μL extract was used in an Agilent 7890B gas chromatograph (Agilent, Palo Alto, CA, USA) coupled with a Time-of-Flight Pegasus HT Mass Spectrometer (Leco, St. Joseph, MI, USA). A Restek Rtx-5Sil MS column (30m, 0.25mm ID, 0.25mm df) (Restek Corp., Bellefonte, PA, USA) was used with flow rate of 1 mL/minute, and 1 μL injection volume. The gradient initial temperature was 50 °C, final temperature of 300 °C, temperature rate of 10 °C/s, and a total time of 20 minutes. The MS ion source was electron ionization with positive mode, mass range of 60−520 m/z, and a source temperature of 250 °C. LC-MS: 200 μL extract was used in a NEXERA XR UPLC system (Shimadzu, Columbia, MD, USA) with a Triple TOF 6600 System (AB Sciex, Framingham, MA, USA). A Phenomenex F5 column was coupled to a Phenomenex F5 pre-column and a pre-column filter with a 0.5 μm A-102 frit (Phenomenex, Torrance, CA, USA). A binary mobile phase system was used with the aqueous phase 100% Optima H2O with 0.1% formic acid (phase A) and the organic phase was 100% acetonitrile (phase B). The gradient was as follows: 0 minutes at 0%B, 4 minutes at 70%B, 8.5 minutes at 100%B, hold until 11.55 minutes, 12 minutes at 0%B, 15 min at 0%B. The flow rate was 0.4000 mL/minute, and the injection volume was 10 μL. For MS, positive and negative polarity was performed separately for each sample. The scan m/z ranges were 100−1200 daltons for TOF MS and 400−600 daltons for product ion scan. The ion source was DuoSpray Ion Source in ESI mode. LC-MS/MS: 150 μL extract was used in a NEXERA XR UPLC system (Shimadzu, Columbia, MD, USA) with a Triple Quadrupole 5500 (AB Sciex, Framingham, MA). An apHera™ NH₂ HPLC column (5 μm, 15 cm × 2 mm) was coupled to an apHera™ NH₂ HPLC guard column (5 μm, 1 cm × 2 mm) and a pre-column filter with a 0.5 μm A-102 frit. A mobile phase binary system was used with aqueous phase 100% Optima H2O with 50mM ammonium hydroxide (phase A) and organic phase was 100% acetonitrile (phase B). The gradient was as follows: 0 minutes at 85%B, 3 minutes at 85%B, 11 minutes at 70%B, 14 minutes at 40%B, 20 minutes at 25%B, 21 minutes at 2%B, hold until 28 minutes, 28.1 minutes at 85%B, end at 29 minutes. The oven temperature was 35 °C, flow rate 0.30 mL/minutes, and injection volume 10 μL.

Metabolites were annotated, known accurate masses and retention times, and in-house authentic standards analysis. The following reference databases were utilized: METLIN, NIST MS, Wiley Registry of Mass Spectral Data, HMDB, MassBank of North America, MassBank Europe, Golm Metabolome Database, SCIEX Accurate Mass Metabolite Spectral Library, MzCloud, and IDEOM. Missing values were imputed by replacement with 1/5 the limit of detection (the minimum recorded value for each metabolite). Data was combined from the three techniques into one data frame, with metabolites measured across more than one technique filtered by priority LC-MS/MS > LC-MS > GC-MS. 475 metabolites were identified, and metabolites with >20% missing values were removed from analyses, leaving 458 quantified metabolites. Metabolite abundances were batch-corrected with ComBat[85], log-transformed, median-centered, and scaled to mean zero, standard deviation one before downstream analyses.

Associations between the abundance of each metabolite abundance and milk CMV status was estimated using a multivariate regression with 'lm' in R. Additional included covariates were the study center (MN vs. OK), parity, maternal age, maternal pre-pregnancy BMI, maternal gestational diabetes status, maternal self-reported race (white vs. non-white) and maternal Healthy Eating Index total score[86] (averaged from three timepoints: prenatal, 1 month postpartum, and 3 months postpartum). P values were corrected for multiple tests using the Benjamini-Hochberg false discovery rate[82] with 'p.adjust' in R v4.3.1[87]. To test for a correlation between CMV-mapped read proportion and metabolite abundance, the same multivariate model was used, replacing CMV status with the CMV-mapped read proportion (logged and scaled to mean 0, s.d. 1) and including only CMV+ samples.

## Infant fecal metagenomics and comparison with milk CMV status

Infant fecal sample collection, DNA extraction, metagenomic sequencing, and estimation of microbial taxon and pathway abundances from 1 and 6 month samples has been previously described[16,48]. Feces were collected from diapers either during study visits and frozen at −80 °C immediately, or collected at home, stored in 2 ml cryovials with 600 μl RNALater (Ambion/Invitrogen, Carlsbad, CA), shipped to the University of Minnesota, and stored at −80 °C. DNA was extracted using the PowerSoil kit (QIAGEN, Germantown, MD. Metagenomic sequencing libraries were generated with the Illumina Nextera XT kit (Illumina, San Diego, CA, United States). Libraries were sequenced with the Illumina NovaSeq system (Illumina, San Diego, CA) with an S4 flow cell and 2 × 150 bp paired-end V4 chemistry at the University of Minnesota Genomics Center to a depth of ~4.5 million reads per sample.

Microbial taxon abundances were estimated by processing metagenomic fastq files with Shi7 version 1.0.1[88]. Sequences were trimmed, filtered by quality scores, and stitched per the learned parameters in Shi7. Sequences were aligned with BURST version 0.99.7f[89], using a reference genome database generated from GTDB r95 (https://gtdb.ecogenomic.org/stats/r95). A 95% identity cutoff and forward/reverse complement flag were used. Resulting.b6 files were converted to reference and taxonomy tables using embalmulate[89] with 'GGtrim' activated. To generate microbial pathway abundances, metagenomic sequences were run through the HUMAnN[90] version 3.0.0 pipeline with MetaPhlAn version 3.0.7, BowTie2[80] version 2.4.2 64-bit, DIAMOND[91] version 0.9.24, and MinPath[92] version 1.5.

Principal components analysis of 1 and 6-month infant metagenomes, summarized as taxon or pathway abundances, was performed separately. Data were filtered to include only taxa/pathways with relative abundance >0.001 in at least 10% of 1-month or 6-month samples, leaving 114 1-month taxa, 100 6-month taxa, 447 1-month pathways, and 469 6-month pathways. A centered log-ratio transformation was performed on the relative abundances of each sample, replacing abundances of zero with a pseudocount of half the minimum non-zero abundance. Principal components were calculated with the 'prcomp' command in R v4.3.1. Associations between the metagenomic PCs that explained at least 5% of the variance in the data (5 PCs each for 1 and 6-month taxa abundances, 3 PCs each for pathway abundances at 1 and 6 months) and milk CMV status were calculated using linear regression with the 'glm' command in R. Infant delivery mode (cesarean vs. vaginal), maternal parity, maternal age, maternal self-identified race, maternal pre-pregnancy BMI, maternal gestational diabetes (yes/no), maternal Group B streptococcus status, fecal sample collection site (home vs. study visit), and maternal Healthy Eating Index total score[49] (averaged from three timepoints: prenatal, 1 month postpartum, and 3 months postpartum) were included as covariates. Two additional covariates were included in the regression models for 6 month infant fecal samples: exclusive breastfeeding status at 6 months (yes/no), and if complementary foods had been introduced at 6 months (yes/no). At 1 month, all infants were exclusively breastfed with no complementary foods. Additional variables about antibiotics use were not included (beyond Group B Streptococcus status, which is treated with antibiotics during labor) because there was too much missing data that would vastly reduce the sample size for these analyses.

Alpha diversity was calculated for each infant fecal sample from 1 or 6 months with the inverse Simpson index with the unfiltered taxon count matrix using the vegan[93] package in R. Alpha diversity was scaled to mean 0, s.d. 1 and tested for association with milk CMV status in a multivariate regression model including the same covariates described above for the microbiome PCs.

Associations between individual taxon abundances and milk CMV status were estimated using a linear mixed effects model with the 'lmerTest' package[94] in R. Using taxon abundances (centered log-transformed and scaled to mean 0, standard deviation 1 within each timepoint) from both 1 and 6-month timepoints as the response variable; fixed effects variables were milk CMV status, sample time-point (1 or 6 months, coded as 0 or 1), infant delivery mode (cesarean or vaginal), maternal parity, maternal self-reported race, maternal pre-pregnancy BMI, maternal Group B streptococcus status, fecal sample collection site (home vs. study visit), maternal gestational diabetes (yes/no), and exclusive breastfeeding status at 6 months; and the mother-infant pair ID was included as a random variable. Only species-level taxa with relative abundance >0.001 in at least 10% of samples in both 1 and 6-month samples were included (56 species). P values were corrected using the Benjamini-Hochberg false discovery rate with 'p.adjust' in R.

### Identification of differentially expressed genes by infant fecal microbial taxon abundances

Correlation of milk gene expression analysis with abundances of CMV-associated infant fecal microbial taxa was performed in DESeq2[49] using the gene-level read count matrix generated with RNA-SeQC[77]. Four differential gene expression analyses were performed: 1 month infant fecal taxon abundances ($N = 104$, 18,817 genes) or 6 month infant fecal taxon abundances ($N = 107$, 18,940 genes), with and without milk CMV status as a covariate. Taxon abundances were centered log-transformed and scaled to mean 0, standard deviation 1 within each timepoint. Additional included covariates were maternal age, maternal pre-pregnancy BMI, maternal self-reported race, maternal parity, infant age in days at study visit, infant delivery mode (cesarean or vaginal), maternal Group B streptococcus status, sample RIN, RNA-sequencing pool, and the sample extracted RNA mass. None of the individuals with transcriptomic data had gestational diabetes, so this was not included as a covariate. P values were adjusted for multiple tests using the default Benjamini and Hochberg method in DESeq2[49,82].

### Infant growth measurement and comparison with milk CMV status

Infant growth measurements and Z-score calculation from this cohort have been previously described[75,95]. Age and sex-specific length-for-age, weight-for-age, and WLZ were calculated using the World Health Organization standards for term infants[50]. Association between infant 1-month WLZ and milk CMV status was calculated in a regression model including WLZ at birth, infant race (parental report), maternal pre-pregnancy BMI, maternal gestational diabetes (yes/no), household income, and delivery mode (cesarean vs. vaginal) as covariates with the 'lm' command in R. Associations between milk CMV status and 3- and 6-month WLZ were calculated in the same model, replacing the outcome (1-month WLZ) with the 3- or 6-month WLZ. Associations with length-for-age or weight-for-age Z-scores used the same covariates, replacing WLZ at birth with the respective Z-score at birth. To test for a correlation between CMV-mapped read proportion and WLZ, CMV status was replaced in the model with the CMV-mapped read proportion (logged and scaled to mean 0, s.d. 1) and including only CMV+ samples.

To further examine the relationships between milk CMV, milk kynurenine, and infant growth, we performed SEM with the R package 'lavaan'[96] version 0.6–17. All models were evaluated with maximum likelihood (ML) parameter estimation with 1000 bootstraps. First, to examine the relationships between milk CMV status, milk kynurenine abundance, and infant growth, we filtered the data to 200 mother-infant pairs with no missing data for four variables (binary milk CMV status, logged and scaled milk kynurenine abundance, and infant WLZ at birth and 1 month). WLZ at birth was included in all models as it is a significant predictor of 1-month WLZ ($r = 0.29$, $P = 2.3 \times 10^{-5}$). Four models (Supplementary Fig. 14) were chosen to test for possible mediation of the relationship between CMV status and infant growth by

milk kynurenine, with (Model 1) or without (Model 2) a direct effect of CMV status on infant growth; or possible mediation of the relationship between milk kynurenine and infant growth by milk CMV status, with (Model 3) or without (Model 4) a direct effect of kynurenine on infant growth. Model fit was evaluated by Chi-squared test ($X^2$ P value > 0.05), comparative fix index (CFI > 0.95), normed fit index (NFI > 0.95), root-mean-square error of approximation (RMSEA < 0.05), and standardized root-mean residuals (SRMR < 0.05). The model that passed all criteria and had the lowest Akaike information criterion (AIC) (Model 3) is highlighted in Supplementary Fig. 13 and Fig. 5D.

Second, to examine the relationships between milk CMV viral load, milk kynurenine abundance, and infant growth, we filtered the data to 76 mother-infant pairs with CMV+ milk and no missing data for four variables (logged and scaled milk proportion CMV-mapped reads as a proxy for viral load, logged and scaled milk kynurenine abundance, and infant WLZ at birth and 1 month). WLZ at birth was included in all models as it is a significant predictor of 1-month WLZ ($r = 0.29$, $P = 2.3 \times 10^{-5}$). Four models (Supplementary Fig. 15) were chosen to test for possible mediation of the relationship between CMV viral load and infant growth by milk kynurenine, with (Model 1) or without (Model 2) a direct effect of CMV viral load on infant growth; or possible mediation of the relationship between milk kynurenine and infant growth by milk CMV viral load, with (Model 3) or without (Model 4) a direct effect of kynurenine on infant growth. We assessed model fit by the same criteria as above. The model that passed all criteria and had the lowest AIC is highlighted in Supplementary Fig. 15 (Model 1) and Fig. 5E.

### Reporting summary

Further information on research design is available in the Nature Portfolio Reporting Summary linked to this article.

## Data availability

Milk RNA and DNA sequencing data have been deposited in dbGaP (https://www.ncbi.nlm.nih.gov/gap/) under study accession phs003408.v1.p1 [https://www.ncbi.nlm.nih.gov/projects/gap/cgi-bin/study.cgi?study_id=phs003408.v1.p1]. Raw sequencing data are available under controlled access in compliance with the study IRB. Use of the data is limited to health/medical/biomedical purposes, including methods development and excluding the study of population origins. Data access is provided by dbGaP (https://dbgap.ncbi.nlm.nih.gov/aa/wga.cgi?page=login) for certified investigators and does not require local IRB approval. Milk and infant fecal metagenomic sequencing data are available at SRA accession PRJNA1019702. Raw metabolomics data is available at Metabolights accession MTBLS10138. Milk metabolite abundances, gene expression matrices, and microbial abundance tables are available in Supplementary Data 14.

## Code availability

The code to reproduce the analyses and figures in this manuscript is available at https://github.com/kelsj/CMV_milk_genomics (https://doi.org/10.5281/zenodo.11224392).

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

## Acknowledgements

We would like to thank Katy Duncan, Laurie Foster, Tipper Gallagher, and all MILk study staff and participants for their contributions, and members of the Albert and Blekhman labs for helpful discussions related to this project. This work was supported by the resources and staff at the University of Minnesota Genomics Center (https://genomics.umn.edu). This work was carried out in part by resources provided by the Minnesota Supercomputing Institute (https://www.msi.umn.edu/). This study was supported by a University of Minnesota Department of Pediatrics Masonic Cross-Departmental Research Grant (F.W.A., R.B., E.W.D., C.A.G.); the University of Minnesota Masonic Children's Hospital Research Fund Award (C.A.G., E.W.D., and D.K.); NIH/NICHD grants R01HD109830 (R.B., E.W.D., C.A.G.), R01HD099866 (M.R.S.), R21HD099473 (C.A.G.), and F32HD105364 (K.E.J.); NIH/NIDCR T90DE0227232 (K.E.J.); a University of Minnesota Office of Academic and Clinical Affairs Faculty Research Development Grant (C.A.G., E.W.D., K.M.J., and D.K.); and a Presbyterian Health Foundation Team Science Award (M.C.R.). The MILK Study, which provided the cohort and milk samples for this study, was supported by NIH/NICHD grant R01HD080444 (E.W.D. and D.A.F.).

## Author contributions

Conceptualization: K.E.J., C.A.G., M.R.S., F.W.A., E.W.D., R.B., formal analysis: K.E.J., T.H., funding acquisition: K.E.J., D.K., K.M.J., E.F.L., D.A.F, C.A.G., F.W.A., E.W.D., R.B., M.R.S., investigation: K.E.J., N.H.A., M.B., T.H., M.A. Supervision: M.C.R., M.R.S., C.A.G., F.W.A., E.W.D., R.B. Writing—original draft: K.E.J. Writing—review and editing: K.E.J., N.H.A., M.B., T.H., M.A., D.A.F., E.I., K.M.J., D.K., E.F.L., M.C.R., C.A.G., M.R.S., F.W.A., E.W.D., R.B.

## Competing interests

The authors declare no competing interests.

## Additional information

¹Department of Genetics, Cell Biology, and Development, University of Minnesota, Minneapolis, MN, USA. ²Department of Pediatrics, University of Minnesota Medical School, Minneapolis, MN, USA. ³Department of Pediatrics, Section of Diabetes and Endocrinology, University of Oklahoma Health Sciences Center, Oklahoma City, OK, USA. ⁴Joslin Diabetes Center, Harvard Medical School, Boston, MA, USA. ⁵Department of Obstetrics, Gynecology and Women's Health, Division of Maternal-Fetal Medicine, University of Minnesota Medical School, Minneapolis, MN, USA. ⁶BioTechnology Institute, College of Biological Sciences, University of Minnesota, Minneapolis, MN, USA. ⁷Department of Computer Science and Engineering, University of Minnesota, Minneapolis, MN, USA. ⁸Division of Biostatistics and Health Data Science, University of Minnesota School of Public Health, Minneapolis, MN, USA. ⁹Harold Hamm Diabetes Center, Department of Biochemistry and Physiology, Oklahoma University Health Sciences Center, Oklahoma City, OK, USA. ¹⁰Division of Epidemiology and Community Health, University of Minnesota School of Public Health, Minneapolis, MN, USA. ¹¹Section of Genetic Medicine, Division of Biological Sciences, University of Chicago, Chicago, IL, USA. ¹²These authors jointly supervised this work: Frank W. Albert, Ellen W. Demerath, Ran Blekhman. ✉e-mail: kej@umn.edu

