## [Peer Review File · Nature Communications]

REVIEWER COMMENTS

Reviewer #1 (Remarks to the Author):

This is a convenience study of the impact of CMV in breast milk on breast milk composition, and the microbiome and anthropometrics of term infants. Data from the MILK study were reanalyzed to classify 276 breast milk samples as CMV+ or CMV- based on the presence of DNA reads mapping to the CMV genome, and comparing the groups with respect to the milk transcriptome and metabolome at 1 month postpartum, as well as to the infant gut microbiome data and growth parameters at 1 and 6 months. The main results are that transcriptomic differences were observed between CMV+ and CMV- milk samples, with CMV+ samples showing an upregulation of interferon signaling pathway genes. There was also an upregulation of the IDO1 gene in CMV+ samples and a corresponding increase in kynurenine and kynurenic acid, which are the products of IDO enzyme conversion of tryptophan. There was no association between detection of CMV in milk at 1 month and microbiome alpha diversity at 1 or 6 months, though CMV positivity was associated with principal components analysis 3 and the abundance of selected bacterial taxa at 1 but not 6 months. Finally, CMV+ milk was associated with greater weight for age at 1 month (but not 6 months), and the number of mapped CMV reads (used as a proxy for viral load) showed the opposite trend.

The study questions are of great interest, since CMV is a highly prevalent infection among women of childbearing age, and whether CMV has any impact on term infants is unclear. Because CMV reactivation is so prevalent (nearly universal) in the breast milk of infected women, it is natural to ask what the consequences might be. CMV infection in utero or in preterm infants can have severe consequences, but until now postnatal CMV infection has only been reported to have negative effects on the growth of HIV-exposed infants in Africa. One study of Canadian infants reported reduced microbial diversity in the stool associated with early CMV infection. However, the conclusions of the study are far from straightforward, and the interpretation is complicated by several significant methodological limitations.

First, the classification of CMV+ and CMV- by shotgun sequence analysis is problematic. We are not told the CMV serostatus of the mother, and DNA is extracted from one time point only. As mentioned, nearly all infected women shed CMV in breast milk though this is often intermittent. Thus, not finding CMV on one day, does not mean that it wasn't there the day before or the one after. Being able to assess maternal infection seems essential. The sensitivity or specificity of the described method for detecting CMV in milk, e.g. compared to standard PCR, is not provided. Even less clear is the fidelity of mapped read abundance with viral load. To make things even more complicated, 2 different DNA extraction methods were used for the parent study, which apparently only both found CMV in 24 samples among the 97 classified as having CMV detected by either method. It is only speculation, but the total positivity rate of 34% would seem low if the population seroprevalence were ~50%. Thus, it is far from clear that the classifications are robust. It is also not

clear which infants became infected with CMV, which might be more influential on their microbiome or growth than exposure to milk.

With respect to the analyses, it is unclear which among them were planned a priori, how much was fishing, and exactly how statistical adjustment for potential multiple confounding overall was handled. For example, was the proportion of immune cells inferred by transcriptome data, which is also of unclear reliability) planned ahead of time, and does a p value of 0.04 really denote a difference given all of the other parameters that were compared?

In the end, even if the data are assumed to be reliable, what is the story? Is increased inflammation the cause or the effect of CMV in milk? Is there any actual nutritional value or other important aspect of the milk. (Other studies have found that correlates of mastitis like sodium concentration or neutrophil count are associated with fat and other macronutrient content.) It is similarly unclear if the microbiome differences are of any potential clinical significance or not, and the anthropomorphic findings, if not erroneous, appear inconsequential.

In summary, this area of research is of considerable interest, but due to its limitations, this particular study does not provide definitive data on whether exposure of health term infants to CMV in breast milk is meaningful or not.

Reviewer #2 (Remarks to the Author):

Thank you for the opportunity to review this paper.

In this manuscript, the authors explored CMV reactivation and this association with human milk composition (RNA sequencing and milk metabolomics) and infant outcomes (microbiome composition and infant growth). Leaning on new and existing phenotypic datasets, the authors identified associations with CMV DNA signatures in human milk. RNA sequencing data revealed elevated gene expression of genes associated with immune cells (based on comparison to publicly available human milk single-cell RNA sequencing data). In this elevated pool of genes, IDO1 is highlighted as it corresponds to two significantly altered human milk metabolites (kynurenine and kynurenic acid) that were detected out of 458 tested metabolites, a finding that the authors acknowledge is published elsewhere. The authors then shifted their focus to infant outcomes. CMV+ status was associated with a selection of Bifidobacterium and a few additional taxa at 1 month and 6 months. CMV+ status was positively associated with infant growth measure (weight-for-length z-

score) and this is likely due to a negative association of length-for-age z score at 1m. Because the trend is not related to viral load, which demonstrates an opposite trend, the authors suggest it may be related to another factor. The authors go on to postulate that this may in part be caused by the elevated kynurenine in milk though they acknowledge issues with this connection in the discussion (e.g. the relationship with kynurenine and WLZ is statistically significant but the effect size may not be clinically relevant, and the authors are not sure whether milk kynurenine levels are present within the infant at high enough levels to have a physiological effect).

While I commend the authors for their integration of multi-omic analyses, I do have questions pertaining to methods, and I would like to see more of a mechanistic connection between what currently are multiple independent observations.

The methods for both the breastmilk metabolic profiles and the infant gut microbiome profiles are not detailed. The authors ask the reader to track down other articles to understand how samples and analyses were handled, but this should be detailed within the manuscript itself.

Based on what I could glean from other publications, here are my questions for the metabolomic analyses.

1. Does the profile of 458 metabolites include all the measured metabolites?
2. What frequency of metabolites were detected in all vs part of the samples?
3. How were metabolites which were detected below the limit of detection treated? Were they imputed?
4. Were some removed? If so, what was the frequency cut off for this removal?
5. What was the unit value for the metabolites?
6. I see that all metabolites were given a pseudo count of 1 before log transformation, but I can imagine that some metabolites would be detected at higher concentrations than others. How was the decision to add a blanket pseudo count of 1 made and did this significantly skew low vs high abundant metabolites?
7. Were all of the samples run in a single batch or were their multiple batches? If there were multiple batches, how was batch effect corrected for?

Of the microbiome sequencing analyses, this was received from the authors

“Feces were collected from diapers either during study visits and frozen at -80°C immediately, or collected at home, stored in 2 ml cryovials with 600 µl RNALater (Ambion/Invitrogen, Carlsbad, CA), and stored at -80°C after shipping to the lab at the University of Minnesota. DNA was extracted using the PowerSoil kit (QIAGEN, Germantown, MD), eluted with 100 µl of the provided elution solution, and stored in microfuge tubes at -80°C.

Extracted DNA was used to construct libraries for metagenomics sequencing using the Illumina Nextera XT ¼ kit (Illumina, San Diego, CA, United States). Metagenomics libraries were then sequenced on an Illumina NovaSeq system (Illumina, San Diego, CA) using the S4 flow cell with the 2x150 bp paired end V4 chemistry kit by the University of Minnesota Genomics Center, achieving a sequencing depth of ~4.5 million reads per sample.

Microbial taxon abundances were generated by first processing metagenomic fastq files with Shi7 version 1.0.191, which learns optimal quality control parameters from the data. Sequences were then trimmed, filtered by quality scores, and stitched per the learned parameters in Shi7. Sequences from all samples were multiplexed into a single fasta file for downstream processing. Processed sequences were aligned to reference databases using BURST version 0.99.7f92, using a reference genome database generated from GTDB r95 (<https://gtdb.ecogenomic.org/stats/r95>). A 95% identity cutoff and forward/reverse complement flag were used. Resulting .b6 files were converted to reference and taxonomy tables using embalmulate92 with ‘GGtrim’ activated. To generate microbial pathway abundances, metagenomic sequences were run through the MetaPhlan93 version 3.0.7 pipeline, with BowTie294 version 2.4.2 64-bit, DIAMOND95 version 0.9.24, and MinPath96 version 1.5.”

1. Please include this within this manuscript as well. Additionally, I am curious how MetaPhlan was used to calculate pathway abundances as this is a tool for species detection. Is it possible this dataset was run through HUMAnN which selects its references based on species identified in MetaPhlan? If so, what reference version was used here?
2. How many species and pathways were retained after prevalence and abundance filters?
3. What pseudocount method was used for CLR transformation?
4. In Figure 4B, are these representatives of aggregated species abundance or a single taxon within that species? I only ask because I see multiple *B. fragilis* and multiple *B. kashiwanohense*? This data has been scaled, but what is the overall abundance of these taxa? What time point were each of these coefficients drawn from?

The connection between CMV and growth is interesting, but there is not much linking this to the other multi-omic analyses. I understand that a similar trend was seen with Kynurenine and this was

postulated in the results section to explain this connection. Did the authors perform a structural equation model to test their hypothesis and justify this statement?

Additionally, were any of the changes within the infant microbiome (e.g. *B. infantis*) also associated with differences in infant growth?

Were any of the changes in the infant microbiome associated with differences in milk RNA expression?

Minor:

The description of the covariates used for each analysis move around. Sometimes they are in both the methods and results section and sometimes they are only in the methods. Please make this more uniform (my preference is to include them all in the results section).

I am curious as to why the authors only included PC analysis for pathways and not a differential analysis for specific pathways like was done for species.

Reviewer #3 (Remarks to the Author):

Review for Nature Communications

The authors investigated in mother-infant pairs who were exclusively breast-fed at the age of 1 months sequencing of milk DNA, RNA and metabolomics and at the age of 1 and 6 months stool DNA sequencing and somatic parameter of the infants.

276 probes were done for DNA sequencing, 216 for RNA sequencing, 176 for metabolomics analyses; 120 fecal analyses and 246 measurements. 96 were CMV positive.

Milk DNA was extracted and sequenced using two approaches for two distinct original goals: low-pass human whole genome sequencing (WGS) or shotgun metagenomic sequencing (SMS). 34 of 36 RNA genes were upregulated in CMV+ milk and the most significantly enriched pathway was the

“cellular response to interferon-gamma”. The proportion of CMV-mapped DNA reads and expression of the differentially expressed genes was significantly positively correlated for two genes: BATF2 and IDO1.

CMV+ milk samples also had a higher estimated proportion of immune cells. Hence, the elevated expression of these genes in CMV+ milk samples stem from an increased proportion of immune cells in CMV+ milk.

Two metabolites were significantly differentially abundant after correcting for multiple tests: kynurenine and its metabolite kynurenic acid. Milk IDO1 expression was positively correlated with the kynurenine/tryptophan ratio of abundances in milk, independent of milk CMV status, thus, illustrating the strong link between expression of IDO1 and the abundance of these metabolites.

Further analyses in short revealed CMV+ milk is associated with decreased Bifidobacterium in the infant gut. Data indicate a complex relationship between milk CMV, milk kynurenine, and infant growth; with kynurenine positively correlated, and CMV viral load negatively correlated, with infant weight-for-length at 1 month of age. These results suggest CMV transmission, CMV-related changes in milk composition, or both may be modulators of full term infant development. However, this difference did not persist to 6 months of age.

This a very well done study with interesting findings indicating a lot of further research to better understand the mechanisms between CMV positive and negative milk and influences on genes expression, metabolomics and somatic growth.

Major concerns:

Please explain the relatively low rate of CMV positive Milk (Kurath et al, Clin Microbiol Infect 2010)

As there are finally hypotheses stated, I would prefer to write “might”

I would be interested in the proportion of mothers who breast-fed until 6 months and whether there were enough dyads to investigate differences correlated with duration of breast-feeding.

The conclusion in the abstract might be changed to a suggestion where further research based on your findings will go to. A “complex relationship” also might be changed to a more conclusive statement regarding findings of the study. For sure it is a complex relationship, but you might find a better description.

Additionally, please discuss why you did not find differences at 6 months or state some hypotheses, which might indicate further investigation. Even your further experiments, shortly mentioned, would be helpful for the interested reader.

Minor concerns: None

Color code:

Reviewer comments

Author's response*Changes to text***REVIEWER COMMENTS*****Reviewer #1 (Remarks to the Author):***

This is a convenience study of the impact of CMV in breast milk on breast milk composition, and the microbiome and anthropometrics of term infants. Data from the MILK study were reanalyzed to classify 276 breast milk samples as CMV+ or CMV- based on the presence of DNA reads mapping to the CMV genome, and comparing the groups with respect to the milk transcriptome and metabolome at 1 month postpartum, as well as to the infant gut microbiome data and growth parameters at 1 and 6 months. The main results are that transcriptomic differences were observed between CMV+ and CMV- milk samples, with CMV+ samples showing an upregulation of interferon signaling pathway genes. There was also an upregulation of the IDO1 gene in CMV+ samples and a corresponding increase in kynurenine and kynurenic acid, which are the products of IDO enzyme conversion of tryptophan. There was no association between detection of CMV in milk at 1 month and microbiome alpha diversity at 1 or 6 months, though CMV positivity was associated with principal components analysis 3 and the abundance of selected bacterial taxa at 1 but not 6 months. Finally, CMV+ milk was associated with greater weight for age at 1 month (but not 6 months), and the number of mapped CMV reads (used as a proxy for viral load) showed the opposite trend.

The study questions are of great interest, since CMV is a highly prevalent infection among women of childbearing age, and whether CMV has any impact on term infants is unclear. Because CMV reactivation is so prevalent (nearly universal) in the breast milk of infected women, it is natural to ask what the consequences might be. CMV infection in utero or in preterm infants can have severe consequences, but until now postnatal CMV infection has only been reported to have negative effects on the growth of HIV-exposed infants in Africa. One study of Canadian infants reported reduced microbial diversity in the stool associated with early CMV infection. However, the conclusions of the study are far from straightforward, and the interpretation is complicated by several significant methodological limitations.

First, the classification of CMV+ and CMV- by shotgun sequence analysis is problematic. We are not told the CMV serostatus of the mother, and DNA is extracted from one time point only. As mentioned, nearly all infected women shed CMV in breast milk though this is often intermittent. Thus, not finding CMV on one day, does not mean that it wasn't there the day before or the one after. Being able to assess maternal infection seems essential. The sensitivity or specificity of the described method for detecting CMV in milk, e.g. compared to standard PCR, is not provided. Even less clear is the fidelity of mapped read abundance with viral load. To make things even more complicated, 2 different DNA extraction methods were used for the parent study, which apparently only both found CMV in 24 samples among the 97 classified as having CMV detected by either method. It is only speculation, but the total positivity rate of 34% would seem low if the population seroprevalence were ~50%. Thus, it is far from clear that the classifications are robust. It is also not clear which infants became infected with CMV, which might be more influential on their microbiome or growth than exposure to milk.

We appreciate the reviewer's concern about the robustness of our approach of detecting CMV in breast milk. To address the reviewer's comments about the sensitivity and specificity of our approach mapping shotgun DNA reads to the CMV genome, we utilized additional aliquots from the same milk samples to measure CMV DNA via qPCR. We found strong agreement between CMV status and viral load estimated using the shotgun sequencing and qPCR approaches (**new Figure 1E-F and Supplementary Fig. 4**). Using the qPCR CMV data, we also repeated the associations between milk CMV status and milk gene expression, milk kynurenine, the infant gut microbiome, and infant growth (additional columns in **Supplementary Table 2**, pasted below). Below we have pasted the new text and **new Supplementary Fig. 4** describing this in the Results section "Identifying CMV-positive samples from shotgun DNA sequencing of human milk":

*To validate our approach of identifying CMV+ milk samples from shotgun sequencing data, we utilized qPCR to detect CMV DNA in a subset of 187 of the same milk samples by an established protocol^{33,34}. The shotgun sequencing and qPCR results were in strong agreement (**Figure 1E, Supplementary Fig. 4, Supplementary Table 1**). Taking qPCR detection of CMV as ground truth, the shotgun sequencing approach had 92.7% sensitivity (95% CI: 92.5-92.9%) and 94.7% specificity (95% C.I. 94.6-94.8%) to identify CMV+ samples (**Supplementary Table 1**). Conversely, taking the shotgun data as ground truth, qPCR detection had 87.9% sensitivity (95% CI: 87.7-88.2%) and 96.9% specificity (95% CI: 96.8-97.0%). Within milk samples identified as CMV+ by both approaches, the qPCR viral load estimate was highly correlated with the proportion of mapped reads from shotgun data (Pearson's $r = 0.88$, $P=3.3 \times 10^{-17}$; **Figure 1F**). As shotgun sequencing data was available for a larger sample, and all major results were consistent when using only the qPCR data (**Supplementary Table 2**), we moved forward with our CMV status designations from mapping shotgun reads to the CMV genome.*

Supplementary Table 2, new columns with qPCR results highlighted in yellow:

	Shotgun sequencing data				qPCR	Shotgun + qPCR
	0% quantile	10% quantile	25% quantile	50% quantile	NA	0% quantile
CMV status						
Minimum proportion CMV-aligned reads	2.70E-08	8.80E-08	3.40E-07	1.50E-06	NA	2.70E-08
Number of CMV+, CMV- samples	97, 187	87, 187	73, 187	49, 187	26, 41	127, 224
Milk gene expression						
Number of significant genes	36	26	37	30	9	37
IDO1 log2 fold change	1.96	2.1	2.27	2.56	1.82	2.02
IDO1 P-value	1.62E-15	3.69E-16	2.00E-17	5.23E-16	7.51E-11	5.15E-17
Milk metabolites						
Number of significant metabolites	2	2	2	2	3	2
Kynurenine effect estimate	0.742	0.837	0.926	0.997	0.826	0.756
Kynurenine P-value	2.30E-06	2.52E-07	7.31E-08	1.67E-07	2.35E-06	1.05E-08
Infant fecal microbiome						
1-month PC3 effect estimate	1.79	1.76	1.84	1.64	1.99	1.82
1-month PC3 P-value	0.00109	0.0026	0.00379	0.0232	0.00715	0.00128
B. infantis effect estimate	-0.507	-0.515	-0.558	-0.578	-0.377	-0.328
B. infantis P-value	0.0006	0.0013	0.0015	0.0043	0.0411	0.0204
Infant growth (1-month)						
WLZ ~ CMV status, effect estimate	0.353	0.296	0.246	0.0882	0.265	0.329
WLZ ~ CMV status, P-value	0.0113	0.0414	0.101	0.622	0.106	0.0105
WLZ ~ CMV load, effect estimate	-0.19865	-0.21957	-0.20328	-0.2112	-0.30917	-0.16596
WLZ ~ CMV load, P-value	1.12E-03	4.46E-03	0.0224	0.176	2.59E-04	8.32E-04

Supplementary Figure 4. Comparison of CMV detection in milk samples using shotgun sequencing data vs. qPCR. Left side column shows a confusion matrix of the count of samples detected as CMV+ vs. CMV- by each method. Right hand column shows the correlation between estimated viral load by qPCR vs. proportion of shotgun reads mapped to the CMV genome. Correlation coefficients were calculated using only estimated viral load for samples that were CMV+ by both qPCR and shotgun sequencing. Viral load estimates are plotted at log-10 scale on both axes. Each row represents a different shotgun data source (see Methods): **(A-B)** all shotgun data combined, $N=187$. In **(B)** samples are colored by the source of shotgun sequencing data. **(C-D)** only shotgun metagenomic sequencing data, SMS, $N=65$; **(E-F)** only whole genome sequencing data, WGS, $N=171$.

We also now discuss these new results and possible reasons for the discrepancy between the 35% positivity rate of our study and the population seropositivity rate in the Discussion:

*We utilized shotgun DNA sequencing from the cell pellet of human milk to identify samples with the presence of CMV at 1 month postpartum, and validated by qPCR. Our study demonstrates that non targeted DNA sequencing of human milk can be used to identify CMV+ samples. We identified CMV DNA in 35% of 1 month milk samples by this approach, which is lower than the estimated seroprevalence for US women of childbearing age (~60%, but highly variable by geographic location and demography)^{35,50,51}. As serum samples were not collected, the seropositivity rate of our study is unknown; however, the demographic characteristics of our cohort (mostly white and highly educated; **Table 3**) indicate the rate is likely lower than the national average^{35,50,51}. Virtually all seropositive women will have CMV reactivation in the mammary gland during lactation⁵², and CMV viral loads are estimated to peak around 4-6 weeks postpartum^{5,8}, the approximate time of sampling for this study. While we may have been unable to detect CMV in some samples with a low viral load, the strong agreement between our shotgun sequencing and qPCR approaches suggests we robustly identified those samples with detectable CMV DNA. We also acknowledge that while viral reactivation during lactation is likely the primary cause of CMV DNA in breast milk, CMV could also be shed through breast milk in the context of primary infections or re-infections occurring late in gestation.*

With respect to the analyses, it is unclear which among them were planned a priori, how much was fishing, and exactly how statistical adjustment for potential multiple confounding overall was handled. For example, was the proportion of immune cells inferred by transcriptome data, which is also of unclear reliability) planned ahead of time, and does a p value of 0.04 really denote a difference given all of the other parameters that were compared?

We appreciate the Reviewer's concern about the statistical robustness of our study. We first tested for associations between CMV and milk composition (metabolites and gene expression), and given that we found a relationship there we thought it would be interesting to then test for associations with infant traits. We corrected for multiple tests using a Benjamini-Hochberg false discovery rate threshold within each genomics analysis, and report results with FDR < 0.05, following standard practice. We did not pre-register our analyses, and have added this statement to the Methods section "*Description of study population*":

The genomics analyses described in this study were not pre-registered.

Regarding the specific example given by the reviewer, the association with cell type proportions, we performed this analysis after finding that there were differences in milk cell gene expression associated with CMV. As the associated genes were enriched for immune response pathways, this was a complementary analysis to that finding. These are not independent analyses in that the cell type proportions are estimated from the gene expression data, and thus we did not perform multiple test correction. We have added additional explanation to Results section "*Immune response genes are upregulated in CMV+ milk samples*":

We also note that the estimated immune cell proportion is not independent of the differential gene expression results above, as both analyses utilize the same bulk RNA-sequencing data.

In the end, even if the data are assumed to be reliable, what is the story? Is increased inflammation the cause or the effect of CMV in milk? Is there any actual nutritional value or other important aspect of the milk. (Other studies have found that correlates of mastitis like sodium concentration or neutrophil count are associated with fat and other macronutrient content.) It is similarly unclear if the microbiome differences are of any potential clinical significance or not, and the anthropomorphic findings, if not erroneous, appear inconsequential.

In summary, this area of research is of considerable interest, but due to its limitations, this particular study does not provide definitive data on whether exposure of health term infants to CMV in breast milk is meaningful or not.

We thank the reviewer for this note. We agree that our study is observational, and cannot make a definitive statement on whether the signature of inflammation in milk is the cause or effect of CMV reactivation. Nevertheless, we believe it is a valuable contribution to the field: it is the first study to test milk compositional differences in a validated measure of CMV reactivation among full term infants. Moreover, the finding showing significant, specific differences between CMV+ and CMV- milk is highly valuable knowledge both for mammary biology and epidemiological studies of infant outcomes. Acquisition of CMV through breast milk is not inconsequential for extremely low-birth-weight premature infants, who not only experience considerable short-term morbidity upon acquisition of CMV by this route, but may be at risk for long-term neurodevelopmental and audiological consequences as well (Weimer et al. 2020). Importantly, the immunomodulatory effects conferred by CMV infections acquired by this route, although they may not confer obvious immediate consequence to term babies, could predispose the premature infants to an increased risk of secondary bacterial and fungal infections, and to an increased risk for chronic lung disease (Hernandez-Alvarado et al. 2021). We highlight these points in the Discussion:

Given our observational study design, we cannot determine if the association with increased IDO1/kynurenine is a cause or consequence of mammary CMV reactivation. Overall, the impact of CMV on milk composition was notably narrow, with a handful of genes and two metabolites differentially abundant between CMV+ and CMV- milk samples. ... Whether kynurenine metabolites in milk are at high enough concentrations to have physiological effects in the infant, and the potential impacts of CMV on this pathway, are possible areas of future investigation.

...

Further studies are required to characterize the impacts of CMV+ milk on growth and the gut microbiome in infants with and without CMV transmission, including in vulnerable preterm infants who most strongly benefit from receipt of human milk but also are at risk for short and long-term health complications upon acquisition of CMV by this route^{3,60}.

Further, while we agree that the observed differences in infant outcomes by CMV status may not immediately change clinical decision-making, they make a novel contribution to the growing field of milk genomic analysis and highlight sources of normal human variation in breast milk. This is similar to other results in the infant microbiome field; for example, studies that first showed that cesarean vs. vaginal delivery was associated with differences in the infant gut microbiome spurred increased fundamental research documenting factors shaping the development of the infant gut microbiome even though in many cases, cesarean delivery is still indicated. We strongly believe that there is value in reporting these patterns, which can now be further explored and tested in future work. We have included the following statement in the Discussion section to make clear this limitations:

We cannot infer causality of the effects of CMV in milk on infant traits due to our observational study design, but these patterns can now be further explored and tested.

Reviewer #2 (Remarks to the Author):

Thank you for the opportunity to review this paper.

In this manuscript, the authors explored CMV reactivation and this association with human milk composition (RNA sequencing and milk metabolomics) and infant outcomes (microbiome composition and infant growth). Leaning on new and existing phenotypic datasets, the authors identified associations with CMV DNA signatures in human milk. RNA sequencing data revealed elevated gene expression of genes associated with immune cells (based on comparison to publicly available human milk single-cell RNA sequencing data). In this elevated pool of genes, IDO1 is highlighted as it corresponds to two significantly altered human milk metabolites (kynurenine and kynurenic acid) that were detected out of 458 tested metabolites, a finding that the authors acknowledge is published elsewhere. The authors then shifted their focus to infant outcomes. CMV+ status was associated with a selection of Bifidobacterium and a few additional taxa at 1 month and 6 months. CMV+ status was positively associated with infant growth measure (weight-for-length z-score) and this is likely due to a negative association of length-for-age z score at 1m. Because the trend is not related to viral load, which demonstrates an opposite trend, the authors suggest it may be related to another factor. The authors go on to postulate that this may in part be caused by the elevated kynurenine in milk though they acknowledge issues with this connection in the discussion (e.g. the relationship with kynurenine and WLZ is statistically significant but the effect size may not be clinically relevant, and the authors are not sure whether milk kynurenine levels are present within the infant at high enough levels to have a physiological effect).

While I commend the authors for their integration of multi-omic analyses, I do have questions pertaining to methods, and I would like to see more of a mechanistic connection between what currently are multiple independent observations.

We would like to thank the Reviewer for their interest in our study, positive comments, and helpful suggestions.

The methods for both the breastmilk metabolic profiles and the infant gut microbiome profiles are not detailed. The authors ask the reader to track down other articles to understand how samples and analyses were handled, but this should be detailed within the manuscript itself.

We thank the reviewer for catching this omission. We have updated the Methods section “*Human milk metabolomics and identification of differentially abundant metabolites*” with a much more detailed description:

Samples for milk metabolomics were prepared and analyzed as previously described⁸² from frozen milk samples at BPGbio Laboratory (Framingham, MA). 200 μ L aliquots of 1-month milk samples were thawed, and metabolites were extracted and processed by three analysis techniques: gas chromatography combined with high-resolution TOF MS, reversed-phase liquid chromatography coupled with high-resolution MS, and hydrophilic interaction chromatography with LC-MS/MS. Samples were processed in 10 batches, with 10 pooled milk samples and 40 external standards included to assess batch-to-batch variability. 475 metabolites were identified, and metabolites with more than 20% missing values were removed from analyses, leaving 458 quantified metabolites. Missing values were imputed by replacement with 1/5 the limit of detection (the minimum recorded value for each metabolite). Metabolite abundances were median-centered, log transformed, and scaled to mean zero, standard deviation one before downstream analyses.

Based on what I could glean from other publications, here are my questions for the metabolomic analyses.

1. Does the profile of 458 metabolites include all the measured metabolites?

2. What frequency of metabolites were detected in all vs part of the samples?
3. How were metabolites which were detected below the limit of detection treated? Were they imputed?
4. Were some removed? If so, what was the frequency cut off for this removal?

We have now included the answers to the Reviewer's questions 1-4 above about the metabolomics pipeline in the Methods section "*Human milk metabolomics and identification of differentially abundant metabolites*", and pasted below:

475 metabolites were identified, and metabolites with more than 20% missing values were removed from analyses, leaving 458 quantified metabolites. Missing values were imputed by replacement with 1/5 the limit of detection (the minimum recorded value for each metabolite).

5. What was the unit value for the metabolites?

As this metabolomics approach is semi-quantitative, there is no absolute unit value but rather an abundance which we then log-transformed. We have added this explanation to the Methods section "*Human milk metabolomics and identification of differentially abundant metabolites*":

Metabolite abundances were batch-corrected with ComBat⁸⁴, median-centered, log transformed, and scaled to mean zero, standard deviation one before downstream analyses.

6. I see that all metabolites were given a pseudo count of 1 before log transformation, but I can imagine that some metabolites would be detected at higher concentrations than others. How was the decision to add a blanket pseudo count of 1 made and did this significantly skew low vs high abundant metabolites?

We thank the reviewer for noticing this, which was a remnant from an earlier version of the analysis of metabolite data that had not had missing values imputed. All metabolite analyses are now log transformed without adding a pseudocount, with no change to the results. We have corrected the Methods section "*Human milk metabolomics and identification of differentially abundant metabolites*" to reflect that we now do not use a pseudocount in the metabolite analyses:

Metabolite abundances were batch-corrected with ComBat⁸⁴, median-centered, log transformed, and scaled to mean zero, standard deviation one before downstream analyses.

7. Were all of the samples run in a single batch or were their multiple batches? If there were multiple batches, how was batch effect corrected for?

Metabolites were run in multiple batches, we have added the following to the Methods section "*Human milk metabolomics and identification of differentially abundant metabolites*":

Samples were processed in 10 batches of 35 samples each, with 10 pooled milk samples and 40 external standards included to assess batch-to-batch variability.... Metabolite abundances were batch-corrected with ComBat⁸⁴

Of the microbiome sequencing analyses, this was received from the authors

“Feces were collected from diapers either during study visits and frozen at -80°C immediately, or collected at home, stored in 2 ml cryovials with 600 µl RNALater (Ambion/Invitrogen, Carlsbad, CA), and stored at -80°C after shipping to the lab at the University of Minnesota. DNA was extracted using the PowerSoil kit (QIAGEN, Germantown, MD), eluted with 100 µl of the provided elution solution, and stored in microfuge tubes at -80°C.

Extracted DNA was used to construct libraries for metagenomics sequencing using the Illumina Nextera XT ¼ kit (Illumina, San Diego, CA, United States). Metagenomics libraries were then sequenced on an Illumina NovaSeq system (Illumina, San Diego, CA) using the S4 flow cell with the 2x150 bp paired end V4 chemistry kit by the University of Minnesota Genomics Center, achieving a sequencing depth of ~4.5 million reads per sample.

Microbial taxon abundances were generated by first processing metagenomic fastq files with Shi7 version 1.0.191, which learns optimal quality control parameters from the data. Sequences were then trimmed, filtered by quality scores, and stitched per the learned parameters in Shi7. Sequences from all samples were multiplexed into a single fasta file for downstream processing. Processed sequences were aligned to reference databases using BURST version 0.99.7f92, using a reference genome database generated from GTDB r95 (<https://gtdb.ecogenomic.org/stats/r95>). A 95% identity cutoff and forward/reverse complement flag were used. Resulting .b6 files were converted to reference and taxonomy tables using embalmulate92 with ‘GGtrim’ activated. To generate microbial pathway abundances, metagenomic sequences were run through the MetaPhlan93 version 3.0.7 pipeline, with BowTie294 version 2.4.2 64-bit, DIAMOND95 version 0.9.24, and MinPath96 version 1.5.”

1. Please include this within this manuscript as well.

We have now added this information to the Methods section “*Infant fecal metagenomics and comparison with milk CMV status*”:

Infant fecal sample collection, DNA extraction, metagenomic sequencing, and estimation of microbial taxon and pathway abundances from 1 and 6 month samples has been previously described^{16,48}. Feces were collected from diapers either during study visits and frozen at -80°C immediately, or collected at home, stored in 2 ml cryovials with 600 µl RNALater (Ambion/Invitrogen, Carlsbad, CA), shipped to the University of Minnesota, and stored at -80°C. DNA was extracted using the PowerSoil kit (QIAGEN, Germantown, MD). Metagenomic sequencing libraries were generated with the Illumina Nextera XT kit (Illumina, San Diego, CA, United States). Libraries were sequenced with the Illumina NovaSeq system (Illumina, San Diego, CA) with anhe S4 flow cell and 2x150 bp paired end V4 chemistry at the University of Minnesota Genomics Center to a depth of ~4.5 million reads per sample.

Microbial taxon abundances were estimated by processing metagenomic fastq files with Shi7 version 1.0.186. Sequences were trimmed, filtered by quality scores, and stitched per the learned parameters in Shi7. Sequences were aligned with BURST version 0.99.7f87, using a reference genome database generated from GTDB r95 (<https://gtdb.ecogenomic.org/stats/r95>). A 95% identity cutoff and forward/reverse complement flag were used. Resulting .b6 files were converted to reference and taxonomy tables using embalmulate88 with ‘GGtrim’ activated. To generate microbial pathway abundances, metagenomic sequences were run through the HUMAnN⁸⁹ version 3.0.0 pipeline with MetaPhlan version 3.0.7, BowTie2⁷⁷ version 2.4.2 64-bit, DIAMOND⁹¹ version 0.9.24, and MinPath^{72,92} version 1.5.

Additionally, I am curious how MetaPhlAn was used to calculate pathway abundances as this is a tool for species detection. Is it possible this dataset was run through HUMAnN which selects its references based on species identified in MetaPhlAn? If so, what reference version was used here?

We thank the reviewer for noticing this; indeed, we have used HUMAnN for pathway abundance quantification, which uses MetaPhlAn as part of the pipeline. We have corrected the Methods section to the following:

To generate microbial pathway abundances, metagenomic sequences were run through the HUMAnN⁸⁹ version 3.0.0 pipeline with MetaPhlAn version 3.0.7, BowTie2⁷⁷ version 2.4.2 64-bit, DIAMOND⁹¹ version 0.9.24, and MinPath^{72,92} version 1.5.

2. How many species and pathways were retained after prevalence and abundance filters?

We have now included this information in the Methods section “*Infant fecal metagenomics and comparison with milk CMV status*”:

Data were filtered to include only taxa/pathways with relative abundance >0.001 in at least 10% of 1-month or 6-month samples, leaving 114 1-month taxa, 100 6-month taxa, 447 1-month pathways, and 469 6-month pathways.

3. What pseudocount method was used for CLR transformation?

We have now included this information in the Methods section “*Infant fecal metagenomics and comparison with milk CMV status*”:

A centered log-ratio transformation was performed on the relative abundances of each sample, replacing abundances of zero with a pseudocount of half the minimum non-zero abundance.

4. In Figure 4B, are these representatives of aggregated species abundance or a single taxon within that species? I only ask because I see multiple *B. fragilis* and multiple *B. kashiwanohense*?

Thanks to the reviewer for this comment, and we now realize the species names used in this figure can be confusing. The taxon designations are from the reference genome database used (GTDB v95). For example, *B. fragilis* A is designated as a separate species from *B. fragilis* by GTDB (English et al. 2023). We have added the following to the Figure 4 legend to make this more clear:

Taxon names ending in ‘A’ were identified as distinct species clusters by sequence identity in the reference genome database (see Methods).

This data has been scaled, but what is the overall abundance of these taxa? What time point were each of these coefficients drawn from?

Figure 4B shows the effect estimate of CMV+ milk on species abundances from a linear mixed model of both timepoints. We agree this was confusing in the original caption and have edited the figure legend to the following to make this more clear. We have also augmented **Supplementary Table 10** to show the mean relative abundance of microbial species included in our analyses for infants fed CMV+ vs. CMV- milk at both 1

and 6 months of age, along with the estimated effect of CMV+ milk at each time point separately and the linear mixed model including both timepoints (what is plotted in Figure 4B, below).

From Figure 4 revised legend:

(B) Estimated effect of CMV+ milk on normalized microbial taxa abundances in the infant gut, modeling samples from both 1 and 6 months of age in a linear mixed model with infant age as a covariate (Methods).

The connection between CMV and growth is interesting, but there is not much linking this to the other multi-omic analyses. I understand that a similar trend was seen with Kynurenine and this was postulated in the results section to explain this connection. Did the authors perform a structural equation model to test their hypothesis and justify this statement?

We thank the reviewer for this suggestion, and have now added a structural equation model to this analysis. The Reviewer is referring to our conclusion that the difference in kynurenine between CMV+/CMV- milk samples, or a correlated factor, was the cause of the association between milk CMV status and infant 1 month WLZ. We have now performed structural equation modeling (SEM) as suggested by the reviewer to further interrogate these relationships. We have added these results in **new Figure 5E-F** (pasted below) and **new Supplementary Figures 14-15**. SEM confirmed that the relationship between milk CMV status and WLZ is absent when milk kynurenine is included in the model, and there was no evidence of kynurenine acting as a mediator between CMV and WLZ (Fig. 5E below). When modeling CMV viral load within CMV+ dyads, SEM again did not find evidence for kynurenine acting as a mediator between milk CMV viral load and WLZ within CMV+ dyads (Fig. 5F below). We paste the **new Figure 5E-F, Results, and Methods** for these analyses below.

Figure 5. ... (E) Structural equation modeling of the relationship between milk kynurenine, milk CMV status, and infant 1 month WLZ. Arrows next to numbers represent the standardized effect estimates, with asterisks indicating P-values. There was no evidence of a mediating relationship of milk kynurenine between milk CMV status and infant 1 month WLZ, nor CMV status mediating a relationship between milk kynurenine and infant 1 month WLZ. All tested models and their fit measures are shown in **Supplementary Fig. 14**. **(F)** A structural equation model examining the relationships between milk proportion of CMV-mapped reads ('Milk prop. CMV reads', a proxy for viral load, within CMV+ milk samples), milk kynurenine, and infant 1 month WLZ. Arrows next to numbers represent the standardized effect estimates, with asterisks indicating P-values. The best fit model plotted here found evidence for both kynurenine mediating the relationship between viral load and 1 month WLZ, and a direct effect from viral load to 1 month WLZ in the opposite (negative) direction. All tested models and their fit measures are shown in **Supplementary Fig. 15**.

From results section 'Milk CMV viral load is correlated with infant growth':

We further investigated these conclusions through structural equation modeling (Methods). First, when examining the relationships between milk CMV status, milk kynurenine, and infant 1-month WLZ, the best fit model included no significant relationship between milk CMV status and infant WLZ (**Figure 5D**). This model was chosen over a model with kynurenine mediating a relationship between milk CMV status and 1 month WLZ (Model 1 in **Supplementary Fig. 14**). We do not interpret the direction of the relationship between milk kynurenine and milk CMV status in these models, but rather that they support our above conclusion that the correlation between milk CMV status and 1 month WLZ is spurious. Additionally, when modeling the relationship between the proportion of CMV-mapped reads within CMV+ samples, milk kynurenine, and infant 1 month WLZ, the best fit model included both a direct effect from CMV read proportion to 1 month WLZ and a mediated effect through milk kynurenine (**Figure 5E, Supplementary Fig. 15**). Thus, overall SEM supports both a positive relationship between milk kynurenine abundance and infant 1-month WLZ, and a negative relationship between CMV viral load and 1-month WLZ within babies fed CMV+ milk.

From Methods section 'Infant growth measurement and comparison with milk CMV status':

To further examine the relationships between milk CMV, milk kynurenine, and infant growth, we performed structural equation modeling (SEM) with the R package 'lavaan'⁹³. All models were evaluated with maximum likelihood (ML) parameter estimation with 1000 bootstraps. First, to examine the relationships between milk CMV status, milk kynurenine abundance, and infant growth, we filtered the data to 200 mother-infant pairs with no missing data for four variables (binary milk CMV status, logged and scaled milk kynurenine abundance, and infant WLZ at birth and 1 month). WLZ at birth was included in all models as it is a significant predictor of 1 month WLZ ($r=0.29$, $P=2.3 \times 10^{-5}$). Four models (Supplementary Fig. 14) were chosen to test for possible mediation of the relationship between CMV status and infant growth by milk kynurenine, with (Model 1) or without (Model 2) a direct effect of CMV status on infant growth; or possible mediation of the relationship between milk kynurenine and infant growth by milk CMV status, with (Model 3) or without (Model 4) a direct effect of kynurenine on infant growth. Model fit was evaluated by Chi-squared test (X^2 P-value > 0.05), comparative fit index (CFI > 0.95), normed fit index (NFI > 0.95), root-mean-square error of approximation (RMSEA < 0.05), and standardized root-mean residuals (SRMR < 0.05). The model that passed all criteria and had the lowest Akaike information criterion (AIC) (Model 3) is highlighted in Supplementary Fig. 14 and Figure 5D.

Second, to examine the relationships between milk CMV viral load, milk kynurenine abundance, and infant growth, we filtered the data to 76 mother-infant pairs with CMV+ milk and no missing data for four variables (logged and scaled milk proportion CMV-mapped reads as a proxy for viral load, logged and scaled milk kynurenine abundance, and infant WLZ at birth and 1 month). WLZ at birth was included in all models as it is a significant predictor of 1 month WLZ ($r=0.29$, $P=2.3 \times 10^{-5}$). Four models (Supplementary Fig. 15) were chosen to test for possible mediation of the relationship between CMV viral load and infant growth by milk kynurenine, with (Model 1) or without (Model 2) a direct effect of CMV viral load on infant growth; or possible mediation of the relationship between milk kynurenine and infant growth by milk CMV viral load, with (Model 3) or without (Model 4) a direct effect of kynurenine on infant growth. We assessed model fit by the same criteria as above. The model that passed all criteria and had the lowest Akaike information criterion (AIC) is highlighted in Supplementary Fig. 15 (Model 1) and Figure 5E.

Additionally, were any of the changes within the infant microbiome (e.g. *B. infantis*) also associated with differences in infant growth?

Differences in the infant microbiome were not associated with infant growth. We have added the following to the Results section "*Milk CMV viral load is correlated with infant growth*":

1-month WLZ was not correlated with PC3 of the infant fecal microbiome or with any of the microbial species associated with milk CMV status (Supplementary Table 11).

Were any of the changes in the infant microbiome associated with differences in milk RNA expression?

We have now added additional analyses testing for relationships between the CMV-associated infant gut microbes with genes expressed in milk.

Added to Results section "*Milk CMV status is correlated with composition of the infant gut microbiome*":

We next tested to see if CMV-associated microbial species in infants were correlated with CMV-associated changes in the milk metabolome or milk gene expression. Neither infant 1-month

metagenome PC3 nor any CMV-associated microbial species was associated with milk kynurenine or the proportion of CMV-mapped reads (Supplementary Table 11). Several CMV-associated genes expressed in milk were correlated the abundances of CMV-associated microbial taxa, but these milk gene – infant microbe associations were attenuated by adding milk CMV status as a covariate to the regression model (Supplementary Fig. 10, Supplementary Table 12). Thus, we found little evidence for CMV-related changes in milk composition leading to the observed differences in the infant fecal microbiome.

Added to Methods section “*Identification of differentially expressed genes by infant fecal microbial taxon abundances*”:

Correlation of milk gene expression analysis with abundances of CMV-associated infant fecal microbial taxa was performed in DESeq2⁷⁹ using the gene-level read count matrix generated with RNA-SeQC⁸⁰. Four differential gene expression analyses were performed: 1 month infant fecal taxon abundances (N=104, 18,817 genes) or 6 month infant fecal taxon abundances (N=107, 18,940 genes), with and without milk CMV status as a covariate. Taxon abundances were centered log-transformed and scaled to mean 0, standard deviation 1 within each timepoint. Additional included covariates were maternal age, maternal pre-pregnancy BMI, maternal self-reported race, maternal parity, infant age in days at study visit, infant delivery mode (cesarean or vaginal), maternal Group B streptococcus status, sample RIN, RNA sequencing pool, and the sample extracted RNA mass. None of the individuals with transcriptomic data had gestational diabetes, so this was not included as a covariate. P-values were adjusted for multiple tests using the default Benjamini and Hochberg method in DESeq2^{79,80}.

Minor:

The description of the covariates used for each analysis move around. Sometimes they are in both the methods and results section and sometimes they are only in the methods. Please make this more uniform (my preference is to include them all in the results section).

Thanks for this suggestion, we agree this was confusing in the previous version. We have revised so that now the covariates included are listed in each Results section in addition to the Methods.

I am curious as to why the authors only included PC analysis for pathways and not a differential analysis for specific pathways like was done for species.

We did not test for associations between CMV and specific pathways because there was no correlation with the PCs of the microbial pathways. We have now added this motivation to the text in Results section “*Milk CMV status is correlated with composition of the infant gut microbiome*”:

We did not test for associations between CMV and individual microbial pathways due to the lack of correlation with microbial pathway PCs (Supplementary Table 9).

Reviewer #3 (Remarks to the Author):

Review for Nature Communications

The authors investigated in in mother-infant pairs who were exclusively breast-fed at the age of 1 months sequencing of milk DNA, RNA and metabolomics and at the age of 1 and 6 months stool DNA sequencing and somatic parameter of the infants.

276 probes were done for DNA sequencing, 216 for RNA sequencing, 176 for metabolomics analyses; 120 fecal analyses and 246 measurements. 96 were CMV positive.

Milk DNA was extracted and sequenced using two approaches for two distinct original goals: low-pass human whole genome sequencing (WGS) or shotgun metagenomic sequencing (SMS). 34 of 36 RNA genes were upregulated in CMV+ milk and the most significantly enriched pathway was the “cellular response to interferon-gamma”. The proportion of CMV-mapped DNA reads and expression of the differentially expressed genes was significantly positively correlated for two genes: BATF2 and IDO1.

CMV+ milk samples also had a higher estimated proportion of immune cells. Hence, the elevated expression of these genes in CMV+ milk samples stem from an increased proportion of immune cells in CMV+ milk.

Two metabolites were significantly differentially abundant after correcting for multiple tests: kynurenine and its metabolite kynurenic acid. Milk IDO1 expression was positively correlated with the kynurenine/tryptophan ratio of abundances in milk, independent of milk CMV status , thus, illustrating the strong link between expression of IDO1 and the abundance of these metabolites.

Further analyses in short revealed CMV+ milk is associated with decreased Bifidobacterium in the infant gut. Data indicate a complex relationship between milk CMV, milk kynurenine, and infant growth; with kynurenine positively correlated, and CMV viral load negatively correlated, with infant weight-for-length at 1 month of age. These results suggest CMV transmission, CMV-related changes in milk composition, or both may be modulators of full term infant development. However, this difference did not persist to 6 months of age.

This a very well done study with interesting findings indicating a lot of further research to better understand the mechanisms between CMV positive and negative milk and influences on genes expression, metabolomics and somatic growth.

We thank the reviewer for their supportive comments and insightful suggestions.

Major concerns:

Please explain the relatively low rate of CMV positive Milk (Kurath et al, Clin Microbiol Infect 2010)

The Reviewer refers to the fact that the CMV positivity rate of milk samples in our study (34%) is much lower than reports of population seropositivity (e.g. 51.6–100% in Kurath et al.) (Kurath et al. 2010). While we unfortunately do not know the seropositivity rate of our cohort as serum samples were not collected, we speculate that it is likely lower than the US average due to the demographics of our cohort. Additionally, in this revision, we added qPCR validation of our shotgun sequencing approach to identify CMV+ milk samples. We found strong agreement between the two methods, providing additional support for our original results (see our first response to Reviewer 1 starting on page 2 above, **new Figure 1E-F**, and **new Supplementary Fig. 4**). We have revised the Discussion to reflect these new results and added discussion of the demographic characteristics of our cohort:

We utilized shotgun DNA sequencing from the cell pellet of human milk to identify samples with the presence of CMV at 1 month postpartum, and validated by qPCR. Our study demonstrates that non targeted DNA sequencing of human milk can be used to identify CMV+ samples. We identified CMV DNA in 32% of 1 month milk samples by this approach, which is lower than the estimated seroprevalence for US women of childbearing age (~60%, but highly variable by geographic location

and demography)^{35,50,51}. As serum samples were not collected, the seropositivity rate of our study is unknown; however, the demographic characteristics of our cohort (mostly white and highly educated; **Table 3**) indicate the rate is likely lower than the national average^{35,50,51}. Virtually all seropositive women will have CMV reactivation in the mammary gland during lactation⁵², and CMV viral loads are estimated to peak around 4-6 weeks postpartum^{5,8}, the approximate time of sampling for this study. While we may have been unable to detect CMV in some samples with a low viral load, the strong agreement between our shotgun sequencing and qPCR approaches suggests we robustly identified those samples with detectable CMV DNA. We also acknowledge that while viral reactivation during lactation is likely the primary cause of CMV DNA in breast milk, CMV could also be shed through breast milk in the context of primary infections or re-infections occurring late in gestation.

As there are finally hypotheses stated, I would prefer to write “might”

If we understand this suggestion correctly, the reviewer suggested that we more carefully word our hypotheses and interpretations. We agree, and in the revised version, we have endeavored to make sure these are worded carefully throughout the manuscript; for example, in the Discussion:

We observed a positive association between milk kynurenine and infant growth at 1 month, with higher milk kynurenine correlated with lower length-for-age and greater weight-for-length Z-scores, suggesting milk kynurenine levels might impact growth in early life independent of CMV status.

I would be interested in the proportion of mothers who breast-fed until 6 months and whether there were enough dyads to investigate differences correlated with duration of breast-feeding.

We agree this is an interesting question, but believe that exploring differences correlated with duration of exclusive breastfeeding (beyond the fact that CMV status was uncorrelated, see Supplementary Table 3) are beyond the scope of the current study. A related question, regarding the relationship of milk hormone and cytokine concentrations to the duration of exclusive breastfeeding, has been previously explored in this cohort (Nagel et al. 2021), but a comprehensive test of metabolomic features as predictors of breastfeeding duration has yet to be conducted.

The conclusion in the abstract might be changed to a suggestion where further research based on your findings will go to. A “complex relationship” also might be changed to a more conclusive statement regarding findings of the study. For sure it is a complex relationship, but you might find a better description.

We have edited this sentence in the abstract, referring to the relationship between CMV, kynurenine, and infant growth, to a more conclusive statement of our findings:

Human cytomegalovirus (CMV) is a highly prevalent herpesvirus that is often transmitted to the neonate via breast milk. Postnatal CMV transmission can have negative health consequences for preterm and immunocompromised infants, but any effects on healthy term infants are thought to be benign. Furthermore, the impact of CMV on the composition of the hundreds of bioactive factors in human milk has not been tested. Here, we utilize a cohort of exclusively breastfeeding full term mother-infant pairs to test for differences in the milk transcriptome and metabolome associated with CMV, and the impact of CMV in breast milk on the infant gut microbiome and infant growth. We find upregulation of the indoleamine 2,3-dioxygenase (IDO) tryptophan-to-kynurenine metabolic pathway in CMV+ milk

samples, and that CMV+ milk is associated with decreased Bifidobacterium in the infant gut. Our data indicate two opposing CMV-associated effects on infant growth; with kynurenine positively correlated, and CMV viral load negatively correlated, with infant weight-for-length at 1 month of age. These results suggest CMV transmission, CMV-related changes in milk composition, or both may be modulators of full term infant development.

Additionally, please discuss why you did not find differences at 6 months or state some hypotheses, which might indicate further investigation. Even your further experiments, shortly mentioned, would be helpful for the interested reader.

We have added the following to the Discussion regarding lack of differences at 6 months:

*We also observed that within infants fed CMV+ milk, higher CMV-mapped read proportion (as a proxy for viral load) was negatively correlated with infant weight-for-length and positively correlated with length-for-age at 1 month of age. The total effect of CMV viral load on 1 month WLZ was estimated at -0.24, of comparable magnitude to the effect of WLZ at birth (0.20; **Figure 5F**). However, the association between milk viral load or kynurenine (measured at 1 month) and infant growth did not persist to 6 months, indicating it did not have long-lasting effect.*

We have also added some additional future directions to the Discussion in response to this Reviewer suggestion:

Further studies are required to characterize the impacts of CMV+ milk on growth and the gut microbiome in infants with and without CMV transmission, including in vulnerable preterm infants who most strongly benefit from receipt of human milk. We cannot infer causality of the effects of CMV in milk on infant traits due to our observational study design, but these patterns can now be further explored and tested. Longitudinal sample collection and tracking of viral transmission would give further insight into the dynamics of CMV in human milk and corresponding impacts on the developing infant gut microbiome and immune system.

Minor concerns: None

References cited in this document

- English, Jamie, Fiona Newberry, Lesley Hoyles, Sheila Patrick, and Linda Stewart. 2023. "Genomic Analyses of *Bacteroides Fragilis*: Subdivisions I and II Represent Distinct Species." *Journal of Medical Microbiology* 72 (11). <https://doi.org/10.1099/jmm.0.001768>.
- Hernandez-Alvarado, Nelmary, Ryan Shanley, Mark R. Schleiss, Jensina Ericksen, Jenna Wassenaar, Lulua Webo, Katherine Bodin, Katelyn Parsons, and Erin A. Osterholm. 2021. "Clinical, Virologic and Immunologic Correlates of Breast Milk Acquired Cytomegalovirus (CMV) Infections in Very Low Birth Weight (VLBW) Infants in a Newborn Intensive Care Unit (NICU) Setting." *Viruses* 13 (10). <https://doi.org/10.3390/v13101897>.
- Kurath, S., G. Halwachs-Baumann, W. Müller, and B. Resch. 2010. "Transmission of Cytomegalovirus via Breast Milk to the Prematurely Born Infant: A Systematic Review." *Clinical Microbiology and Infection: The Official Publication of the European Society of Clinical Microbiology and Infectious Diseases* 16 (8): 1172–78.
- Nagel, Emily M., Leslie Kummer, David R. Jacobs Jr., Laurie Foster, Katy Duncan, Kelsey Johnson, Lisa Harnack, et al. 2021. "Human Milk Glucose, Leptin, and Insulin Predict Cessation of Full Breastfeeding and Initiation of Formula Use." *Breastfeeding Medicine: The Official Journal of the Academy of Breastfeeding Medicine* 16 (12): 978–86.
- Weimer, Kristin E. D., Matthew S. Kelly, Sallie R. Permar, Reese H. Clark, and Rachel G. Greenberg. 2020. "Association of Adverse Hearing, Growth, and Discharge Age Outcomes With Postnatal Cytomegalovirus Infection in Infants With Very Low Birth Weight." *JAMA Pediatrics* 174 (2): 133–40.

REVIEWERS' COMMENTS

Reviewer #1 (Remarks to the Author):

The authors have addressed all of my previous comments and the revised manuscript, with its additional data and qualified interpretations, is greatly improved.

Reviewer #2 (Remarks to the Author):

Thank you. All of my questions and concerns have been appropriately addressed.

Reviewer #3 (Remarks to the Author):

My concerns have been reasonably addressed in the revisions, thus, the manuscript might be acceptable.